

# Adiabatic deformations of quantum Hall droplets

## Blagoje Oblak[1,2]⋆ and Benoit Estienne[2]†

**1** CPHT, CNRS, École polytechnique, Institut Polytechnique de Paris, 91120 Palaiseau, France
**2** Laboratoire de Physique Théorique et Hautes Energies,
Sorbonne Université and CNRS UMR 7589, F-75005 Paris, France

⋆ blagoje.oblak@polytechnique.edu , † estiennne@lpthe.jussieu.fr

## Abstract

We consider area-preserving deformations of the plane, acting on electronic wave functions through 'quantomorphisms' that change both the underlying metric and the confining potential. We show that adiabatic sequences of such transformations produce Berry phases that can be written in closed form in terms of the many-body current and density, even in the presence of interactions. For a large class of deformations that generalize squeezing and shearing, the leading piece of the phase is a super-extensive Aharonov-Bohm term ($\propto N^2$ for $N$ electrons) in the thermodynamic limit. Its gauge-invariant subleading partner only measures the current, whose dominant contribution to the phase stems from a jump at the edge in the limit of strong magnetic fields. This results in a finite Berry curvature per unit area, reminiscent of the Hall viscosity. We show that the latter is in fact included in our formalism, bypassing its standard derivation on a torus and suggesting realistic experimental setups for its observation in quantum simulators.

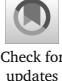

# 1 Introduction and motivation

One of the defining characteristics of topological phases of matter is their robustness under smooth changes of the Hamiltonian. This notably includes geometric deformations of the sample supporting the system; after all, the low-energy field theory of any such phase is topological, hence insensitive to bulk diffeomorphisms [1–3]. Crucially, topological invariance entails the presence of chiral edge modes propagating along the boundary of the system—and these generally *do* react to deformations in a non-trivial manner, one that actually determines the low-energy physics [4–6]. It is therefore of interest to model the effects of geometric deformations on topological phases of matter, including their boundary.

The present work provides just such an analysis in the paradigmatic case of the quantum Hall (QH) effect [7,8]. In that context, the importance of diffeomorphisms has been recognized since the early days, as the symplectic structure of position operators projected to the lowest Landau level is essential both for magneto-rotons [9] and for the non-commutative approach to QH physics [10]. The key actors for these lines of thought are *area-preserving deformations* spanning the Girvin-MacDonald-Platzmann algebra [9], *i.e.* a $w_{1+\infty}$ algebra [11–18]. In particular, a staple of the seminal works [12–14] was the sketch of a relation between bulk deformations and conformal transformations of gapless edge modes. We shall partly rely on these insights to focus on area-preserving maps of physical interest, which we dub 'edge deformations' for reasons that will become clear below (see fig. 1).

An apparently unrelated class of deformations leads to the *Hall viscosity* [19, 20], also

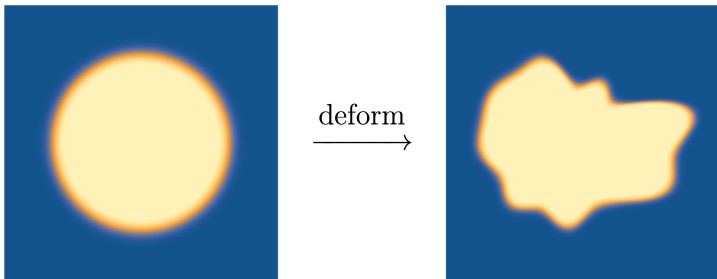

Figure 1: An initially isotropic planar electron droplet deformed by a typical edge deformation (applied using eq. (25) below). The shape of the edge changes in an arbitrary manner while preserving the droplet's area, even in the thermodynamic limit.

known as 'odd viscosity' [21] or 'Lorentz shear modulus' [22, 23], where a QH sample on a torus reacts to adiabatic changes of the modular parameter. In that case, one reads off the Berry curvature [24] associated with linear reparametrizations of the metric, finding an extensive result (proportional to the number $N \gg 1$ of electrons) whose coefficient is quantized and robust against disorder [25], similarly to the Hall conductance [26]. The resulting geometric response and relation to hydrodynamics has been studied in great detail in recent years [27–37], but concrete observations seem elusive despite recent encouraging results in graphene [38]. As we shall show here, the Hall viscosity can in fact be derived on a *plane*, as a response to a simple subset of the aforementioned edge deformations. This bypasses the complicated toroidal wave functions typically needed in standard computations of the Hall viscosity, suggesting that the latter can be measured as a response to specific time-dependent perturbations in tabletop quantum simulators [39–43].

Our approach is based on a one-body formalism and is then extended to many-body droplets. Concretely, we consider area-preserving deformations of the metric and potential in a Landau Hamiltonian (see eq. (37) below). This is done in a gauge-invariant and unitary manner through so-called *quantomorphisms* that may be seen as local generalizations of magnetic translations [44, 45]. One can then choose a set of slow time-dependent deformations, forcing the Hamiltonian to become time-dependent as well. Provided these deformations start and end at the same configuration, and assuming the initial wave function was an energy eigenstate, the final wave function coincides with the initial one up to an uninteresting dynamical phase and a crucial Berry phase [24, 46, 47]. The latter can in fact be written as a fairly simple expectation value when parameter variations are unitary [48–50], as will be the case here. Indeed, we show in this way that adiabatic quantomorphisms produce Berry phases containing two separately gauge-invariant terms: the first is a contribution of the current that appears universally whenever diffeomorphisms act on wave functions, and the second is an Aharonov-Bohm (AB) phase weighted by the density [51]. Schematically,

$$\text{Berry phase} = \int \mathrm{d}^2\mathbf{x}\left(\text{current} \times \text{velocity} + \text{density} \times \text{AB phase}\right), \tag{1}$$

where $\mathrm{d}^2\mathbf{x} = \mathrm{d}x\,\mathrm{d}y$ in Cartesian coordinate, and we refer to eq. (50) below for the detailed expression. We stress that this can all be written in terms of explicit formulas, applying both to one-body states and fully-fledged droplets of $N \gg 1$ electrons, interacting or not.

While the computations that we carry out hold for any charged quantum state in the plane, a regime of particular interest is that of weak potentials and strong magnetic fields. The one-body energy spectrum then splits into familiar Landau levels, resulting in a many-body density that is roughly constant and quantized in the bulk of a droplet, but zero outside. The many-body current, on the other hand, is typically small in the bulk but jumps in a Gaussian fashion near the edge (see fig. 10 below) [52, 53]. This entails a distinction between the two pieces of the Berry phase (1): the AB phase is sensitive to bulk deformations, while the current measures both bulk and edge effects. We eventually illustrate this by restricting attention to a specific class of planar diffeomorphisms, namely the aforementioned 'edge deformations' that contain, in particular, all linear maps

$$\begin{pmatrix} x \\ y \end{pmatrix} \mapsto \begin{pmatrix} a & b \\ c & d \end{pmatrix}\begin{pmatrix} x \\ y \end{pmatrix}, \qquad \text{with} \qquad ad - bc = 1. \tag{2}$$

Acting on a Hall droplet with an adiabatic sequence of such transformations produces a Berry phase (1) whose AB term is super-extensive ($\propto N^2$ for $N$ electrons). Discarding the latter leaves out the gauge-invariant current contribution, which turns out to be extensive in the limit of strong magnetic fields. The corresponding Berry curvature per unit area provides an infinite-parameter analogue of the Hall viscosity, to which it reduces up to an overall factor 2

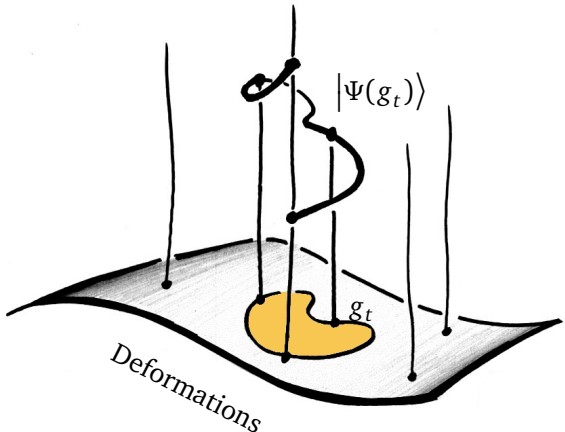

Figure 2: A base space, consisting of all area-preserving planar deformations, supports a line bundle (so the fibers are complex planes). Given some deformation $g$, the fiber above it is the ray of some quantum state $|\Psi(g)\rangle$. The bundle is endowed with a Berry connection. Parallel transport of a quantum state along some closed loop $g_t$ generally results in a net holonomy—a Berry phase. The latter measures an 'area' (highlighted in yellow) in the group of deformations. Adapted from [50, fig. 1].

in the case of linear maps (2) applied to integer QH states. For *fractional* QH states, the Berry curvature that we find is again very similar to Hall viscosity, but differs from it by a factor $2\nu$ in terms of the filling fraction $\nu < 1$. We eventually show that this mismatch is due to the fact that the current term in (1) measures variations of both metric *and* potential, while the Hall viscosity is defined as a response to metric variations alone. In this sense, there was no reason for the Berry phase (1) to be related to viscosity at all; it just so happens that its extensive piece is proportional to the Hall viscosity.[1]

As usual, statements on Berry phases can be phrased either in terms of finite holonomies as in fig. 2, or in terms of local Berry curvatures and linear response.[2] In the case at hand, connections and curvatures are differential forms on an infinite-dimensional parameter space consisting of all area-preserving diffeomorphisms, so they are somewhat unwieldy. We have therefore chosen to present most of our results in terms of (conceptually simpler) finite phases, omitting for now the detailed study of infinite-dimensional quantum geometry. The latter will be crucial in practice, as the likeliest avenue to observe the effects of the phase (1) is to study adiabatic linear response in quantum simulators, where the high degree of control over microscopic details may help overcome issues of decoherence and disorder.

The remainder of this work is organized as follows. We begin with preliminary material in sec. 2, showing first how unitary group actions give rise to parameter-dependent Hamiltonians, then applying the idea to adiabatic diffeomorphisms of one-dimensional (1D) quantum wires. Sec. 3 then introduces area-preserving diffeomorphisms and their unitary, gauge-invariant action on charged wave functions through quantomorphisms [44, 45]. We present this concept in detail and show that diffeomorphisms are represented in a projective manner, with a specific central extension that we compute. Our key results are then derived in sec. 4. This includes the formula (50) for Berry phases of 2D droplets subjected to adiabatic quantomorphisms, its value (63) in the special case of edge deformations, and its application to QH states and

---

[1]The proportionality could have been guessed on geometric grounds: the parameter space for linear maps (2) is a hyperbolic plane, which has a unique $SL(2,\mathbb{R})$-invariant Berry curvature up to normalization.

[2]Holonomies actually contain more information, as they also know about Berry phases along non-contractible cycles—think *e.g.* of the two cycles of a flat torus. In the case of planar deformations of isotropic states, parameter space is homotopic to a point so this subtlety plays no role.

the Hall viscosity (79). Finally, the conclusion is devoted to a brief discussion of follow-ups, while app. A provides a detailed review of the geometry of quantomorphisms in the case of *compact* surfaces such as a torus, where subtleties occur that do not affect the plane. App. B is a technical aside useful for sec. 4.

One last minor warning before we proceed: this work involves tools in differential geometry that are not reviewed in detail. We therefore refer to [54, chaps. 1–2] for useful background on diffeomorphisms, vector fields and flows, and to [55, part I] or [56] for a pedagogical introduction to Lie groups and symplectic geometry. Finally, various aspects of groups of diffeomorphisms, along with some symplectic geometry, are reviewed in [57, chaps. 5–6] in a language that may be closer to the habits of physicists.

## 2 Adiabatic deformations of quantum wires

This section provides crucial background for later use. Specifically, we start by showing that unitary Lie group actions give rise to geometric phases [24, 48] when some group elements fail to commute with the Hamiltonian. This general fact is then applied to diffeomorphisms of wave functions on a circle, with potential implications in quantum wires with persistent currents [58–60]. The resulting phases mimic those produced by adiabatic conformal maps [50], and will eventually appear on the edge of Hall droplets in sec. 4.4.

### 2.1 Berry phases from group actions

Here we sketch a general derivation of Berry phases associated with adiabatic group actions in quantum mechanics, obtained upon seeing group elements as labels for parameter-dependent Hamiltonians [50, sec. 2].

**Parameter-dependence from group theory.** We are interested in transformations that act on a quantum system and span a continuous group. Accordingly, let $G$ be a (connected) Lie group acting on some Hilbert space $\mathcal{H}$, with $\mathcal{U}[g]$ the unitary operator implementing the transformation $g \in G$. We assume that the assignment $g \to \mathcal{U}[g]$ furnishes a representation of $G$, *i.e.* it is compatible with multiplication in $G$; for later convenience we choose a *right* representation, so

$$\mathcal{U}[f]\mathcal{U}[g] = \mathcal{U}[gf], \tag{3}$$

for all $f, g \in G$. We stress that this is merely a matter of convention: it makes some formulas look simpler, but it does not affect the physics.

Now let $H$ be an 'unperturbed' Hamiltonian operator, and force the system to undergo a transformation due to $g \in G$; imagine *e.g.* a spin in a rotating magnetic field, where $g$ is a rotation. Then the transformed Hamiltonian is

$$H[g] = \mathcal{U}[g]H\mathcal{U}[g]^\dagger. \tag{4}$$

If some $\mathcal{U}[g]$'s fail to commute with $H$, the operators $H[g]$ are 'deformed' Hamiltonians, each depending parametrically on a point $g \in G$. Their energy spectrum coincides with that of $H$: $|\Psi\rangle \in \mathcal{H}$ is an eigenstate of $H$ if and only if $\mathcal{U}[g]|\Psi\rangle$ is an eigenstate of $H[g]$ with the same eigenvalue. Importantly, eigenstates depend on $g$ even though their energy does not, which is ultimately why Berry phases appear in this context. Examples include the usual action of 3D rotations on a qubit [24, 61], or that of Lorentz boosts on relativistic states [62–64], or conformal maps in conformal field theory [50]. Starting in sec. 3, $G$ will consist of area-preserving deformations acting on fermionic wave functions.

**Berry phases.** Parameter-dependence entails the presence of geometric phases [24]. In the case at hand, let $g_t$ be some path in $G$ where $t \in [0, T]$ is a time variable, giving rise to a time-dependent Hamiltonian $\mathcal{U}[g_t] H \mathcal{U}[g_t]^\dagger$. The problem is to solve the ensuing Schrödinger equation. If the path $g_t$ is traced very slowly,[3] and provided the initial state vector is an eigenstate of $\mathcal{U}[g_0] H \mathcal{U}[g_0]^\dagger$ with isolated and non-degenerate energy,[4] the solution of the Schrödinger equation is itself a time-dependent energy eigenstate. In formulas, if $|\Psi\rangle$ is a (time-independent) normalized eigenstate of $H$ with energy $E$ and the initial condition is $\mathcal{U}[g_0]|\Psi\rangle$, then the adiabatic theorem [46,65] ensures that the state vector at time $t$ is

$$|\psi(t)\rangle \sim \exp\left[-i\frac{Et}{\hbar} - \int_0^t \mathrm{d}\tau \, \langle\Psi|\mathcal{U}[g_\tau]^\dagger \partial_\tau \mathcal{U}[g_\tau]|\Psi\rangle\right] \mathcal{U}[g_t]|\Psi\rangle, \tag{5}$$

up to corrections that vanish as the rate of change of $g_t$ goes to zero. It follows that any *closed* path $g_t$ in $G$, with period $T$ say, leads to a final state vector $|\psi(T)\rangle \sim e^{i\theta}|\psi(0)\rangle$ that coincides with the initial one up to a phase $\theta$. The latter is the sum of a dynamical phase $-ET/\hbar$ and a *Berry phase* [24,48,50]

$$\mathcal{B}_\Psi[g_t] = i \oint \mathrm{d}t \, \langle\Psi|\mathcal{U}[g_t]^\dagger \partial_t \mathcal{U}[g_t]|\Psi\rangle = -\oint \mathrm{d}t \, \langle\Psi|\mathfrak{u}\big[(\partial_t g_t)g_t^{-1}\big]|\Psi\rangle, \tag{6}$$

where the second equality follows from eq. (3) and $\mathfrak{u}[v]$ is the Hermitian operator obtained by differentiating $\mathcal{U}[g]$ at the identity, for any Lie algebra element $v$:

$$\mathfrak{u}[v] \equiv -i\partial_\epsilon\big|_0 \mathcal{U}[e^{\epsilon v}], \qquad \text{for any} \quad v \in \mathfrak{g}. \tag{7}$$

The phase (6) is thus a functional of the curve $g_t$ and depends parametrically on the reference state $|\Psi\rangle$. Famous examples of this kind include the Berry phases of a spin in a rotating magnetic field [24,61] and Thomas precession [62–64], respectively stemming from unitary rotations and Poincaré transformations.

Note that transformations $g_t$ that belong to the stabilizer of $|\Psi\rangle$ in the sense that $\mathcal{U}[g_t]|\Psi\rangle \propto |\Psi\rangle$ give rise to vanishing phases (modulo $2\pi$). As a result, the actual parameter space responsible for the phases (6) is not quite the group manifold $G$, but its quotient by the stabilizer of $H$. For example, many of the cases treated below will involve isotropic Hamiltonians whose eigenstates have definite angular momentum; these are invariant under rotations, so their parameter space will be a quotient $G/\mathrm{U}(1)$. This subtlety is not crucial in practice, as one can just compute Berry phases and notice, after the fact, that they vanish for certain families of deformations (namely those in the stabilizer).

**Central extensions.** To conclude this general group-theoretic presentation, let us provide a generalization of eq. (6) that will be of the utmost importance later. Suppose indeed that the unitary operators $\mathcal{U}[g]$ furnish a *projective* representation of $G$, *i.e.* that eq. (3) is corrected by a phase factor:

$$\mathcal{U}[f]\mathcal{U}[g] = e^{iC(g,f)} \mathcal{U}[gf]. \tag{8}$$

Here $C(g, f)$ is some non-zero real function, known as a *central extension*, that satisfies the following identity in order to preserve the associativity of composition:[5]

$$C(g, f) + C(h, gf) = C(h, g) + C(hg, f). \tag{9}$$

---

[3]'Slowly' normally means that the operator norm of $\hbar\partial_t(\mathcal{U}[g_t]H\mathcal{U}[g_t]^\dagger)$ is much smaller than the square of the energy gap separating $|\Psi\rangle$ from other eigenstates, but this is too strong since the adiabatic theorem even holds for *gapless* ground states [65]. Indeed, this is the situation we'll encounter in sec. 4.

[4]Non-degeneracy eventually implies that the Berry connection is Abelian; degenerate, non-Abelian cases are not treated here. In sec. 4, the degeneracy of Landau levels will be lifted by a weak confining potential.

[5]Eq. (9) says that $C$ is a two-cocycle in the sense of group cohomology; see *e.g.* [57, sec. 2.1] for details.

Then the second equality in the phase (6) no longer holds, since it relied on the condition that $\mathcal{U}$ be an *exact* representation (with $\mathsf{C} = 0$ in (8)). Instead, when $\mathcal{U}$ is projective, the cocycle $\mathsf{C}$ adds an extra term to the Berry phase: assuming without loss of generality that the neutral element $e \in G$ acts trivially (*i.e.* $\mathcal{U}(e) = \mathbb{I}$) so that $\mathsf{C}(e, g) = 0$, one finds

$$\mathcal{B}_\Psi[g_t] = -\oint dt \, \langle\Psi|\mathfrak{u}[(\partial_t g_t)g_t^{-1}]|\Psi\rangle - \oint dt \, \partial_\tau\big|_t \mathsf{C}(g_\tau, g_t^{-1}). \tag{10}$$

Such an extension will affect area-preserving maps and their Berry phases in secs. 3–4, essentially due to non-commuting magnetic translations.

## 2.2 Berry phases from circle deformations

Having reviewed how unitary group actions yield geometric phases, we now turn to the phases produced by adiabatic deformations of a 1D quantum system on a circle. This is a key toy model for future reference: first because the unitary action of quantomorphisms in sec. 3.3 is inspired by the 1D setup, and second because deformations of the edge of Hall droplets effectively produce 1D phases in sec. 4.3. Note that our discussion of diffeomorphisms of the circle is kept to a minimum; we refer *e.g.* to [57, sec. 6.1] for a smoother introduction. See also [50] for a derivation of Berry phases nearly identical to the ones studied here, albeit in the context of 1D conformal field theory.

**Unitary circle diffeomorphisms.** Consider a particle confined to a 1D ring, *i.e.* a unit circle $S^1$. Let its quantum state be described by a $2\pi$-periodic wave function $\Psi(\varphi)$ in the Hilbert space $L^2(S^1)$. We wish to deform this wave function via diffeomorphisms of the circle, *i.e.* invertible smooth maps $g : S^1 \to S^1$ whose inverse $g^{-1}$ is also smooth. Since we are ultimately interested in continuous paths of diffeomorphisms connected to the identity, we focus on orientation-preserving maps; these span a group denoted $\mathrm{Diff}\, S^1$, with a 'multiplication' given by the composition of functions. The simplest way to describe such maps is to lift them to $\mathbb{R}$, yielding smooth functions $g : \mathbb{R} \to \mathbb{R} : \varphi \mapsto g(\varphi)$ that satisfy $g'(\varphi) > 0$ and $g(\varphi + 2\pi) = g(\varphi) + 2\pi$ for all $\varphi$.[6] For instance, rotations lift to $g(\varphi) = \varphi + \theta$; more general diffeomorphisms can be seen as 'wiggly' versions of rotations (see fig. 3). Other examples of circle diffeomorphisms are given in eq. (21) below.

How should circle deformations act on wave functions? A straightforward answer is provided by the fact that wave functions are half-densities—their norm squared is a measure. Thus a natural action of $\mathrm{Diff}\, S^1$ on $L^2(S^1)$ is

$$\big(\mathcal{U}[g]\Psi\big)(\varphi) \equiv \sqrt{g'(\varphi)}\, \Psi(g(\varphi)), \tag{11}$$

for any $g \in \mathrm{Diff}\, S^1$, where the square root on the right-hand side ensures unitarity.[7] One readily verifies that this is indeed a right representation in the sense of eq. (3). Intuitively, it mimics the fact that deformations of wave functions are induced by those of a sample, and that the local height of a wave function follows the local density of points; see fig. 4 for a cartoon. One can also deduce from (11) the expression of Lie algebra operators (7) that implement infinitesimal diffeomorphisms, *i.e.* vector fields. Indeed, writing $g(\varphi) = \varphi + \epsilon v(\varphi)$ with a $2\pi$-periodic function $v(\varphi)$ and expanding (11) up to first order in $\epsilon$ yields

$$\mathfrak{u}[v]\Psi \overset{(7)}{=} -i \lim_{\epsilon \to 0} \tfrac{1}{\epsilon}(\mathcal{U}[g]\Psi - \Psi) = -i\big(v(\varphi)\partial_\varphi + \tfrac{1}{2}v'(\varphi)\big)\Psi(\varphi). \tag{12}$$

---

[6]The lift is not unique since $g(\varphi)$ and $g(\varphi) + 2\pi n$ describe the same diffeomorphism for any integer $n$, but this ambiguity is harmless since it does not affect space-time derivatives of time-dependent deformations.

[7]Thus deformations are *not* mere changes of variables $\Psi \to \Psi \circ g$, which would not be unitary!

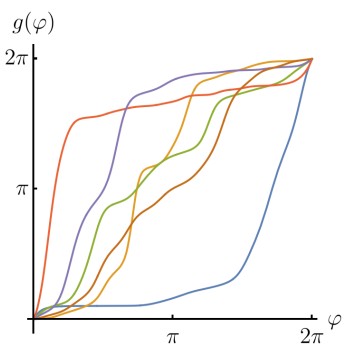

Figure 3: Several generic diffeomorphisms of the circle, all chosen to fix the origin so that $g(0) = 0$ and $g(2\pi) = 2\pi$. Loosely speaking, any such function can be seen as an identity map $g(\varphi) = \varphi$ with extra 'wiggles' (The sharp version of this statement is that any orientation-preserving diffeomorphism of the circle is homotopic to the identity, *i.e.* that the group $\text{Diff}\, S^1$ is connected).

This will be useful to apply the Berry phase formula (6) to time-dependent diffeomorphisms. Note that the operator (12) can be recast in terms of standard position and momentum: using $p = -(i\hbar/R)\partial_\varphi$ on a circle of radius $R$, one has $(\hbar/R)\mathfrak{u}[v] = v(\varphi)p - i(\hbar/2R)v'(\varphi)$, where the second term is a 'correction' ensuring that $\mathfrak{u}[v]$ is Hermitian. One can thus think of $\mathfrak{u}[v]$ as a position-dependent translation, as should indeed be the case for diffeomorphisms.

As explained in sec. 2.1, group elements acting on a reference Hamiltonian $H$ produce deformed Hamiltonians $\mathcal{U}[g]H\mathcal{U}[g]^\dagger$ that depend parametrically on $g$. To develop some intuition on the operators (11), it helps to ask how they actually modify a typical one-body Hamiltonian with some $2\pi$-periodic potential $V$,

$$H = \frac{p^2}{2M} + V(\varphi). \tag{13}$$

Using the definition (11) and working again on a circle of radius $R$, one finds

$$\mathcal{U}[g]H\mathcal{U}[g]^\dagger = \frac{1}{2M}\,p\Big(\frac{1}{g'(\varphi)^2}\Big)p + V\big(g(\varphi)\big) + \frac{\hbar^2}{4MR^2\,g'(\varphi)^2}\Big(\frac{g'''}{g'} - \frac{5}{2}\Big(\frac{g''}{g'}\Big)^2\Big). \tag{14}$$

Here the deformed kinetic term involves a non-Euclidean metric $g'(\varphi)^2\mathrm{d}\varphi^2$, as was to be expected; the ordering makes it manifest that the Hamiltonian is Hermitian. As for the potential term, it is deformed from $\varphi$ to $g(\varphi)$, but also receives an 'anomalous' potential contribution reminiscent of the Schwarzian derivative [66, sec. 4].

By the way, note that the deformation-dependent Hamiltonians (14) suffice for the derivation of Berry phases, without requiring any digression on unitarity. An experimenter could indeed start from the operators (14) and make $g_t$ slowly time-dependent, which would eventually lead to Berry phases in the usual way. It just so happens that the $g$-dependence of wave functions is given, in the present case, by unitary operators (11).

**Berry phases.** Unitary diffeomorphisms (11) give rise to Hamiltonians (14) labelled parametrically by a point $g$ in the $\text{Diff}\, S^1$ group manifold. Following sec. 2.1, one may then study Berry phases resulting from adiabatic, cyclic parameter variations. Let therefore $g_t$ be some closed path in $\text{Diff}\, S^1$, and pick some normalized eigenstate $\Psi$ (with non-degenerate, isolated energy) of the undeformed Hamiltonian $H$. Adiabatic time evolution takes the form (5) with

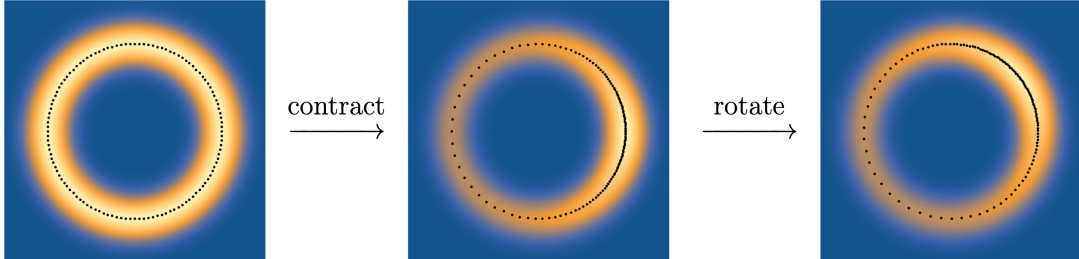

Figure 4: A circle (black, dashed points) supports a wave function with initially uniform density (blue-yellow density plot). The circle is acted upon by two deformations: the first is a local contraction towards $\varphi = 0$ (*i.e.* a dilation away from $\varphi = \pi$); the second is a rotation $\varphi \mapsto \varphi + \theta$. The effect on the wave function is manifest: the initial contraction affects its density through the square root in (11), while the rotation rigidly rotates its argument according to the change of argument in (11). Note that the plot shows the procedure one would actually carry out in a lab: if $g_1$, $g_2$ are the two transformations acting on the wave function according to eq. (11), then the transformations of the circle that are actually shown are the inverses $g_1^{-1}$ and $g_2^{-1}$. This minor subtlety is due to our choice to use *right* group representations (3), and would not occur if all $g$'s in eq. (11) were replaced by $g^{-1}$'s.

a Berry phase (6) that now reads[8]

$$\mathcal{B}_\Psi[g_t] = i \oint \mathrm{d}t \int_0^{2\pi} \mathrm{d}\varphi \, \sqrt{g'_t(\varphi)} \, \Psi^*(g_t(\varphi)) \partial_t \Big( \sqrt{g'_t(\varphi)} \, \Psi\big(g_t(\varphi)\big) \Big), \tag{15}$$

owing to the definition (11). This can be made explicit by evaluating the time derivative, integrating by parts and changing the integration variable from $\varphi$ to $g_t(\varphi)$:

$$\mathcal{B}_\Psi[g_t] = -\frac{M}{\hbar} \oint \mathrm{d}t \, \mathrm{d}\varphi \, j(\varphi) \, \dot{g}_t\big(g_t^{-1}(\varphi)\big), \tag{16}$$

where the dot denotes a partial time derivative, $M$ is the mass of the particle and $j(\varphi)$ is the probability current of $\Psi$,

$$j \equiv \frac{\hbar}{2Mi} \big( \Psi^* \partial_\varphi \Psi - \Psi \partial_\varphi \Psi^* \big). \tag{17}$$

We stress that eq. (16) is an explicit functional of the path of deformations $g_t(\varphi)$, and otherwise only depends on the state $\Psi$ through its probability current $j(\varphi)$. This is a general phenomenon: as we confirm in sec. 4, Berry phases produced by adiabatic diffeomorphisms measure currents of quantum states. Also note that eq. (16) was derived in a one-body context, but it is equally valid in thermodynamically large systems provided the probability current $j(\varphi)$ is replaced by the many-body current density $J(\varphi)$. One may therefore expect the phases (16), or the corresponding linear response, to be observable in 1D quantum wires with persistent currents [58–60].

Our argument so far was a brute-force computation based on eq. (15), but there exists an equivalent derivation in terms of Lie-algebraic data, owing to the expression on the far right-hand side of eq. (6). Indeed, the group $\mathrm{Diff}\,S^1$ consists of deformations of a circle, so its algebra consists of infinitesimal diffeomorphisms $\varphi \mapsto \varphi + v(\varphi)$, *i.e.* vector fields $v(\varphi)\partial_\varphi$. The Lie algebra element $(\partial_t g_t)g_t^{-1}$ in (6) is thus a time-dependent vector field

$$v_t(\varphi) \equiv \frac{\partial}{\partial \tau} g_\tau\big(g_t^{-1}(\varphi)\big)\big|_{\tau=t} = \dot{g}_t\big(g_t^{-1}(\varphi)\big), \tag{18}$$

---

[8]Throughout this work, holonomies and time integrals over periodic paths are written as $\oint$. By contrast, spatial integrals, be they on a circle or on a plane, are denoted by the symbol $\int$.

whose flow is the one-parameter family of diffeomorphisms $g_t$. In hydrodynamics, $v_t$ would be the actual velocity vector field of the fluid flow $g_t$ (see *e.g.* [67]). Abstractly, one can also think of (18) as the logarithmic time derivative of the path of deformations $g_t(\varphi)$. One can then plug (18) in the Lie algebra operator (12) and use the Berry phase

$$\mathcal{B}_\Psi[g_t] = -\frac{M}{\hbar} \oint \mathrm{d}t\, \mathrm{d}\varphi\, j(\varphi)\, v_t(\varphi)\,, \tag{19}$$

to reproduce eq. (16). This reformulation is no surprise, but it will be worth keeping in mind once we turn to 2D diffeomorphisms.

### 2.3 Examples of adiabatic deformations

Let us exhibit time-dependent diffeomorphisms whose Berry phase (16) takes a manageable form. Consider first the simplest case, namely time-dependent rigid rotations $g_t(\varphi) = \varphi + \theta_t$ such that $\theta_T = \theta_0 + 2\pi n$ for some integer $n$. Then eq. (16) yields

$$\mathcal{B}_\Psi[g_t] = -2\pi n\, \frac{M}{\hbar} \int \mathrm{d}\varphi\, j(\varphi)\,, \tag{20}$$

so the Berry phase measures the average current. In particular, if $\Psi = \frac{1}{\sqrt{2\pi}} e^{is\varphi}$ is a plane wave with integer angular momentum $s$, then the current $j(\varphi) = \hbar s/(2\pi M)$ is a quantized constant and the Berry phase vanishes modulo $2\pi$. This was anticipated below eq. (7): the parameter space of isotropic Hamiltonians is not quite the whole group of diffeomorphisms, but its quotient $\mathrm{Diff}\, S^1/S^1$.

Let us now turn to less elementary diffeomorphisms that will turn out to be crucial for the Hall viscosity in sec. 4.4. Namely, given any positive integer $k$, consider a map $\varphi \mapsto g(\varphi)$ defined by

$$e^{ikg(\varphi)} = \frac{\alpha\, e^{ik\varphi} + \beta}{\beta^* e^{ik\varphi} + \alpha^*}\,, \qquad \text{with} \quad \alpha, \beta \in \mathbb{C} \text{ such that } |\alpha|^2 - |\beta|^2 = 1\,. \tag{21}$$

At fixed $k$, such maps span a group locally isomorphic to SL(2,$\mathbb{R}$). A one-parameter family of deformations of this kind is depicted in fig. 5; they stretch the circle by 'pinching it' at $k$ equally distributed points. What happens when these perturbations become time-dependent and act on a rotation-invariant Hamiltonian? In that case the reference state $\Psi$ is a plane wave and the ensuing Berry phase takes the form (16) with constant $j(\varphi) = \hbar s/(2\pi M)$ in terms of an integer angular momentum $s$. Making $\alpha, \beta$ time-dependent in (21) and using eq. (16) then yields

$$\mathcal{B}_\Psi[g_t] = -\frac{2s}{k} \oint \mathrm{d}t\, \mathrm{Im}(\alpha^* \dot{\alpha} - \beta^* \dot{\beta})\,. \tag{22}$$

This is explicit, but not especially illuminating. A more striking result is obtained thanks to SL(2,$\mathbb{R}$) coordinates $(\lambda, \theta, \chi)$ defined via $\alpha \equiv e^{i(\chi+\theta)} \cosh\lambda$ and $\beta \equiv e^{i(\chi-\theta)} \sinh\lambda$, so that eq. (22) becomes

$$\mathcal{B}_\Psi[g_t] = -\frac{2s}{k} \oint \mathrm{d}t\, \dot{\theta}_t \cosh(2\lambda_t)\,, \tag{23}$$

where the contribution of $\chi_t$ drops out since it merely corresponds to adiabatic rotations of an isotropic system. Note that (23) is nothing but a hyperbolic area written in 'polar coordinates' $(\lambda, \theta)$. Indeed, writing (23) as a surface integral $\mathcal{B} = \int \mathcal{F}$, the corresponding Berry curvature

$$\mathcal{F} = -\frac{4s}{k} \sinh(2\lambda)\, \mathrm{d}\lambda \wedge \mathrm{d}\theta\,, \tag{24}$$

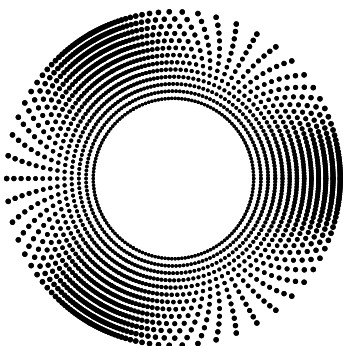

Figure 5: A sequence of diffeomorphisms (21) with $k = 3$ and $(\alpha, \beta) = (\cosh \lambda, \sinh \lambda)$, where $\lambda$ ranges from 0 (innermost circle) to 0.6 (outermost circle). The innermost circle consists of 50 uniformly distributed points; successive deformations spoil uniformity, with manifest density maxima at $\varphi = 0 \mod 2\pi/3$ and minima at $\varphi = \pi/3 \mod 2\pi/3$. (Inner points are smaller than outer ones for readability.) The same kind of deformation, albeit with $k = 1$, was used to produce the contraction in fig. 4.

is proportional to the area form on a hyperbolic plane. This is no coincidence: it lies at the root of both Thomas precession [50, 63, 64] and the Hall viscosity [19, 20], so we return to it in much greater detail in sec. 4.4, where (24) is recast in terms of coordinates $\tau_1, \tau_2$ on an upper half-plane with area form $d\tau_1 \wedge d\tau_2/\tau_2^2$.

## 3 Quantum area-preserving deformations

Having introduced unitary diffeomorphisms in 1D, we now turn to their area-preserving peers in 2D. This is a key preliminary for Berry phases such as (6), which require detailed knowledge of the operators $\mathcal{U}[g]$. Accordingly, this section begins with basic facts on area-preserving diffeomorphisms and their infinitesimal cousins, namely divergence-free ('symplectic') vector fields [56]. We then turn to quantum mechanics and introduce gauge-invariant unitary deformations of wave functions, which are found to coincide with the 'quantomorphisms' normally encountered in geometric quantization [44, 45]. Along the way, we show that the resulting representation is projective in the sense of eq. (8). Numerous examples are provided throughout, notably including 'edge deformations'.

### 3.1 Area-preserving deformations

Area-preserving diffeomorphisms arise naturally in analytical mechanics and symplectic geometry [56, 68], where they are seen as symmetries of phase space. Here we list their basic properties and provide a few examples for future reference, including the key notion of 'edge deformations' inspired by [14]. The presentation is purely classical for now: all quantum aspects are relegated to sec. 3.3.

Consider a plane $\mathbb{R}^2$ supporting a uniform magnetic field $\mathbf{B}$; in Cartesian coordinates $(x, y)$, one can write $\mathbf{B} = B \, dx \wedge dy$ as an area form.[9] By definition, a *diffeomorphism* of the plane is an invertible smooth map $g : \mathbb{R}^2 \to \mathbb{R}^2 : \mathbf{x} \mapsto g(\mathbf{x})$ whose inverse $g^{-1}$ is also smooth. We say that $g$ is *area-preserving* if it leaves the area form invariant ($g^* \mathbf{B} = \mathbf{B}$), *i.e.* if it has unit

---

[9]Bold fonts are used for all objects that carry spatial indices: $\mathbf{x}$ is a position 'vector', $g$ is a 'vector'-valued map, $\mathbf{A}$ is a one-form, $\mathbf{B}$ is a two-form, $\boldsymbol{v}$ is a vector field, etc.

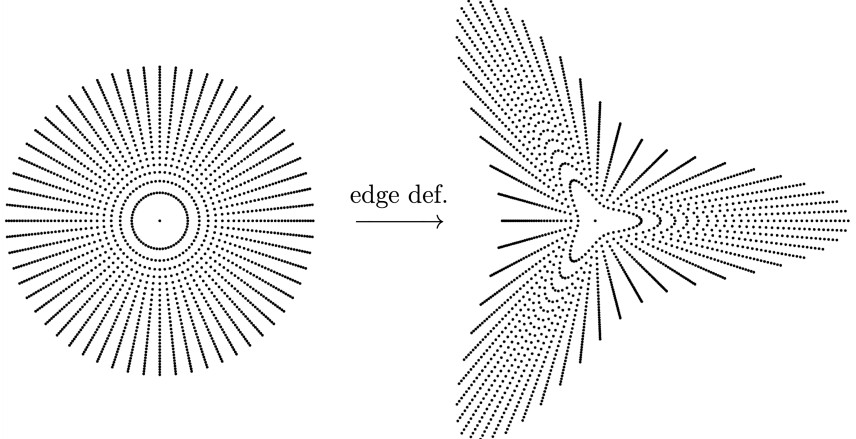

Figure 6: The action of an edge deformation (25) on a disk, with $g(\varphi)$ of the form (21) with $k = 3$, $\alpha = \cosh(1/2)$ and $\beta = \sinh(1/2)$. The 1D diffeomorphism $\varphi \mapsto g(\varphi)$ increases the angular density of points near $\varphi = 0 \mod 2\pi/3$, and decreases it near $\varphi = \pi/3 \mod 2\pi/3$ (recall fig. 5). This local change of angular density is compensated by a modification of radial density: regions where angles contract become radially dilated, and vice-versa.

Jacobian. The set of all such maps is a group under composition, denoted SDiff $\mathbb{R}^2$ for 'special' diffeomorphisms, analogously to the special linear groups SL($n$).

Let us exhibit some classes of diffeomorphisms that will appear below. A first example, somewhat trivial but still important, is given by translations $\mathbf{x} \mapsto \mathbf{x} + \mathbf{a}$; these form an (Abelian) subgroup $\mathbb{R}^2$ of SDiff $\mathbb{R}^2$. Secondly, linear maps (2) have unit Jacobian by definition, and thus span an SL($2, \mathbb{R}$) subgroup of area-preserving maps; this includes an SO(2) subgroup of rotations around the origin. Finally, to introduce an even larger subset of deformations that will later be crucial, consider standard polar coordinates $(r, \varphi)$ defined by $x + iy = r\, e^{i\varphi}$ and ask what area-preserving diffeomorphisms $\boldsymbol{g} : (r, \varphi) \mapsto \boldsymbol{g}(r, \varphi)$ commute with all dilations $(r, \varphi) \mapsto (\lambda r, \varphi)$. The answer is provided by all maps of the form

$$\boldsymbol{g}(r, \varphi) = \left( \frac{r}{\sqrt{g'(\varphi)}}, \; g(\varphi) \right), \tag{25}$$

where $g(\varphi)$ is any (orientation-preserving) circle diffeomorphism in the sense of sec. 2.2. One readily verifies that (25) preserves the magnetic field $\mathbf{B} = B\, r\, dr \wedge d\varphi$: intuitively, any angular 'compression' is compensated by an angle-dependent radial 'dilation' (and vice-versa), as in fig. 6. In fact, one can even make a stronger statement: all deformations (25) leave the symmetric gauge potential $\mathbf{A} = \frac{B}{2} r^2 d\varphi$ invariant since $r^2 d\varphi = \frac{r^2}{g'(\varphi)} dg(\varphi)$.

We shall refer to maps of the form (25) as *edge deformations* because they deform the edge of any isotropic quantum Hall droplet in a non-trivial but finite way regardless of its size (see fig. 1 above). Indeed, (25) sends any circle $r = \text{cst}$ on a deformed curve $r = \text{cst}/\sqrt{g'(\varphi)}$, where $g'(\varphi)$ is independent of $r$. Note that edge deformations span a subgroup of SDiff $\mathbb{R}^2$, isomorphic to the group Diff $S^1$ of circle deformations introduced in sec. 2.2. In fact, their algebra consists of vector fields on a circle, and is thus very similar to the Virasoro algebra of edge modes studied in [14]. Finally, note that the linear maps (2) form a subset of the group of edge deformations, since (2) can be written as (25) in polar coordinates, with a 1D deformation $g(\varphi)$ given by eq. (21) with $k = 2$ and

$$(\alpha, \beta) = \left( \tfrac{1}{2}(a - ib + ic + d), \tfrac{1}{2}(a + ib + ic - d) \right). \tag{26}$$

We will exploit this coincidence at the end of sec. 4.4.

## 3.2 Divergence-free vector fields and incompressible flows

We have seen in eq. (6) that Berry phases produced by unitary transformations can be written in terms of Lie-algebraic data as opposed to finite transformations. In the case of 1D deformations, we exhibited this with eq. (19), involving the 1D vector field (18) and its flow $g_t$. It is therefore essential to become acquainted with the 2D version of these objects, *i.e.* with the Lie algebra of the group $\mathrm{SDiff}\,\mathbb{R}^2$. This will often be useful below, not least because it allows us to introduce stream functions that link the subject to hydrodynamics [69, sec. 4.2] and play an important role in the quantum theory (sec. 3.3).

**Divergence-free vector fields.** The group $\mathrm{SDiff}\,\mathbb{R}^2$ consists of deformations of a plane, so its Lie algebra consists of infinitesimal diffeomorphisms $\mathbf{x} \mapsto \mathbf{x} + \mathbf{v}(\mathbf{x})$, *i.e.* vector fields $\mathbf{v}$. In order to preserve area, these vector fields need to be divergence-free: in terms of Lie derivatives of the magnetic field, one has $\mathcal{L}_{\mathbf{v}}\mathbf{B} = B\,\nabla_i v^i = 0$. For example, infinitesimal edge deformations (25) are generated by vector fields of the form

$$\mathbf{v} = v(\varphi)\partial_\varphi - \frac{r}{2}v'(\varphi)\partial_r \qquad \text{(edge deformations)}, \tag{27}$$

where $v(\varphi)$ is any $2\pi$-periodic function; it is immediate to verify that (27) is indeed divergence-free. Note that the bracket of two divergence-free vector fields is itself divergence-free, so such vector fields spans a Lie algebra, denoted $\mathrm{SVect}\,\mathbb{R}^2$ in analogy with the notation $\mathrm{SDiff}\,\mathbb{R}^2$ for the group of area-preserving deformations. This is actually a $w_{1+\infty}$ algebra [12] whose Witt subalgebra is that of infinitesimal edge deformations (27).

As usual for Lie groups, the map that sends Lie algebra elements on group elements is the exponential. The latter is really a flow in the case of diffeomorphism groups: given a vector field $\mathbf{v}_t$, say even time-dependent, consider the time-dependent diffeomorphisms $\mathbf{g}_t$ given by the 2D version of the logarithmic derivative (18),

$$\mathbf{v}_t = \dot{\mathbf{g}}_t \circ \mathbf{g}_t^{-1}. \tag{28}$$

This family of diffeomorphisms is, by definition, the flow of $\mathbf{v}_t$; in the special case where $\mathbf{v}_t$ is time-independent, $\mathbf{g}_t$ is called the exponential of $\mathbf{v}$. The analogy with hydrodynamics is obvious: one can think of $\mathbf{v}_t$ as the velocity vector field of an incompressible fluid, and $\mathbf{g}_t$ is the resulting flow so that $\mathbf{g}_t(\mathbf{x})$ is the position at time $t$ of a fluid parcel initially at $\mathbf{x}$. Notions of flow will also be important for Berry phases, owing to the appearance of the combination (28) in eq. (10).

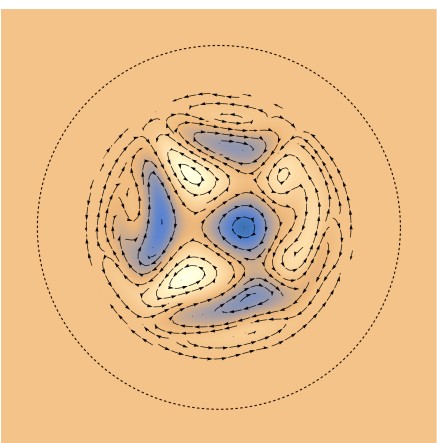

Figure 7: The flow of a divergence-free vector field whose stream function is compactly supported in the bulk of a disk (whose edge is represented by a dashed circle). The density plot shows the stream function, whose level curves are streamlines of the flow. It is clear that such compactly supported functions yield deformations that only affect the bulk without touching the edge. Conversely, one may consider stream functions that only vary in a neighbourhood of the edge, without affecting the bulk.

**Stream functions.** On the plane, any divergence-free vector field is determined by a *stream function*[10] $F$ such that $\iota_v \mathbf{B} = \mathrm{d}F$, which in Cartesian coordinates boils down to

$$v^i = \tfrac{1}{B} \varepsilon^{ij} \partial_j F, \tag{29}$$

where $\varepsilon^{ij}$ is antisymmetric and $\varepsilon^{xy} \equiv +1$. Returning to the examples of sec. 3.1, one would find that the stream functions generating translations $\mathbf{x} \mapsto \mathbf{x} + \mathbf{a}$ and linear transformations (2) are respectively linear and quadratic in Cartesian coordinates $(x, y)$, while the stream function giving rise to the edge vector field (27) is

$$F(\mathbf{x}) = -\frac{1}{2} B r^2 v(\varphi) \qquad \text{(edge deformations)} \tag{30}$$

(One can always add an arbitrary additive constant to a stream function without affecting its vector field; we choose this constant such that $F(0) = 0$ in (30)).

Stream functions highlight the abundance of area-preserving maps, and also provide powerful methods to construct specific deformations with desired properties. For example, any compactly supported stream function $F$ produces a vector field $v$ that vanishes outside of the support; the corresponding flow is then obtained by integrating eq. (28), and consists of diffeomorphisms that only affect the support of $F$ (see fig. 7).

As the terminology suggests, stream functions arise naturally in incompressible hydrodynamics (see *e.g.* [69, sec. 4.2]). But they are also crucial in symplectic geometry, since any area form in 2D is also a symplectic form: the magnetic field $\mathbf{B} = B\,\mathrm{d}x \wedge \mathrm{d}y$ says for instance that the coordinates $x$ and $y$ are canonically conjugate, similarly to 'position' and 'momentum' in 1D mechanics. Stream functions in that context are more commonly called 'Hamiltonian functions' [56], because the corresponding flow (28) coincides with Hamilton's equations of motion with the stream function seen as a Hamiltonian. However, we will steer clear of this

---

[10]This does not hold on any surface! For example, on a torus, translations obviously preserve area but admit no single-valued stream function. The general statement is that all divergence-free vector fields on $\Sigma$ admit a stream function if and only if the first cohomology group $H^1(\Sigma)$ vanishes.

terminology to avoid confusion with the actual Hamiltonian of a quantum system. The symplectic perspective is nevertheless useful thanks to its connection with geometric quantization [44, 45, 70, 71], to which we now turn in order to define unitary deformations of wave functions.

### 3.3 Unitary deformations are quantomorphisms

Here we carry out the 2D version of the 1D construction presented around eqs. (11)–(14). Accordingly, consider a non-relativistic particle with electric charge $q$ in the plane $\mathbb{R}^2$, whose space of states is the usual one-body Hilbert space $L^2(\mathbb{R}^2)$. We wish to act on that particle's wave functions with unitary area-preserving deformations. How to proceed?

The simplest way forward is to recall the 1D definition (11), realize that its square root term involves the Jacobian of the deformation, and adapt the prescription to area-preserving maps (whose Jacobian is trivial):

$$\left(\mathcal{U}[g]\Psi\right)(\mathbf{x}) \stackrel{?}{\equiv} \Psi\left(g(\mathbf{x})\right). \tag{31}$$

This defines a unitary map for any $g \in \text{SDiff}\,\mathbb{R}^2$ and it satisfies $\mathcal{U}[f] \circ \mathcal{U}[g] = \mathcal{U}[g \circ f]$, so it suffices for neutral states or in the absence of magnetic fields. But it suffers from serious problems in the case of charged states: area-preserving maps generally *change* the vector potential $\mathbf{A}$ (*i.e.* $g^*\mathbf{A} \neq \mathbf{A}$) even if they preserve the magnetic field $\mathbf{B} = \mathrm{d}\mathbf{A}$. This implies, on a practical level, that eq. (31) modifies the gauge in which the vector potential is written, preventing the computation of Berry phases which rely on the possibility of comparing phases of wave functions. From a more conceptual standpoint, eq. (31) is plainly ill-defined, in that it presupposes a gauge choice: changing gauges according to $\tilde{\Psi}(\mathbf{x}) \equiv e^{iq\alpha(\mathbf{x})/\hbar}\Psi(\mathbf{x})$ turns (31) into a different representation $\mathcal{U}[g]\tilde{\Psi} = e^{iq(\alpha - \alpha \circ g)/\hbar}\tilde{\Psi} \circ g$. We now show how both of these issues can be fixed; the solution involves quantomorphisms [44, 45], whose formulation in terms of fiber bundles is reviewed in greater detail in app. A.

**Compensating gauge transformations.** We have just pointed out that the operator (31) changes the gauge in which the potential $\mathbf{A}$ is written. To correct this, one can compensate the change of gauge by a pure gauge transformation, *i.e.* define

$$\left(\mathcal{U}[g]\Psi\right)(\mathbf{x}) \stackrel{?}{\equiv} e^{iq\alpha(\mathbf{x})/\hbar}\Psi\left(g(\mathbf{x})\right), \tag{32}$$

with a function $\alpha(\mathbf{x})$ such that $\mathrm{d}\alpha = \mathbf{A} - g^*\mathbf{A}$. Since the one-form $\mathbf{A} - g^*\mathbf{A}$ is closed, and because $\mathbb{R}^2$ is simply connected, $\alpha$ exists globally[11] and can be written as an integral

$$\alpha(\mathbf{x}) = \int_{\mathbf{x}_0}^{\mathbf{x}} (\mathbf{A} - g^*\mathbf{A}). \tag{33}$$

Here the integration contour is any curve going from some 'origin' $\mathbf{x}_0$ to $\mathbf{x}$; its choice is irrelevant since $\mathbf{A} - g^*\mathbf{A}$ is closed. The reference point $\mathbf{x}_0$ is arbitrary and may even depend on $g$, but changing it eventually yields the same operator up to a global phase, so the choice of $\mathbf{x}_0$ is ultimately unimportant as well. In the following, we pick for simplicity the same origin $\mathbf{x}_0 = 0$ for all deformations. With this choice, for instance, eq. (32) applied to any translation $g(\mathbf{x}) = \mathbf{x} + \mathbf{a}$ yields

$$\left(\mathcal{U}[g]\Psi\right)(\mathbf{x}) \equiv e^{iqB(xa_y - ya_x)/2\hbar}\Psi(\mathbf{x} + \mathbf{a}) \qquad \text{(translations)}, \tag{34}$$

when $\mathbf{A} = B(x\,\mathrm{d}y - y\,\mathrm{d}x)/2$ is written in symmetric gauge. This is nothing but a standard (finite) magnetic translation.

---

[11]Similarly to footnote 10, this fails on surfaces with non-trivial first cohomology, *e.g.* the torus.

At this stage we have fixed the main issue: the operators (32) no longer modify the gauge. Furthermore, the definition (32) is *nearly* intrinsic: it takes the same form in all gauge choices, up to a global phase. This last point can be further improved by choosing, for each map $\boldsymbol{g}$, a smooth path $\gamma_{\boldsymbol{g}}$ linking $\mathbf{x}_0$ to $\boldsymbol{g}(\mathbf{x}_0)$ (for instance a straight line). One can then add a global phase to (32) and define

$$\left(\mathcal{U}[\boldsymbol{g}]\Psi\right)(\mathbf{x}) \equiv e^{-iq\int_{\gamma_{\boldsymbol{g}}}\mathbf{A}/\hbar}\, e^{iq\int_{\mathbf{x}_0}^{\mathbf{x}}(\mathbf{A}-\boldsymbol{g}^{*}\mathbf{A})/\hbar}\,\Psi\big(\boldsymbol{g}(\mathbf{x})\big), \tag{35}$$

which is our final prescription for the unitary action of area-preserving deformations on wave functions with charge $q$. Most of the conclusions below ultimately stem from this elementary definition. In the context of geometric quantization, operators (35) are known as *prequantum bundle automorphisms* or *quantomorphisms* [44, 45]. We will adopt the latter terminology, and refer again to app. A for a number of technical details, including the construction of quantomorphisms on surfaces more general than the plane where integrals such as (33) may be globally ill-defined. In particular, we show there that (35) is essentially the *unique* well-defined unitary action of deformations on quantum wave functions.

Recall from sec. 2.2 that 1D diffeomorphisms provide geometry-dependent Hamiltonians (14), involving a deformed metric and potential. In a similar way, the intuitive meaning of quantomorphisms (35) becomes clearer upon acting with some $\mathcal{U}[\boldsymbol{g}]$ on the Hamiltonian

$$H = \frac{1}{2M}(\mathbf{p}-q\mathbf{A})^2 + V(\mathbf{x}), \tag{36}$$

for an electron of mass $M$ in some potential $V$. A brute force computation then yields

$$\mathcal{U}[\boldsymbol{g}]H\mathcal{U}[\boldsymbol{g}]^{\dagger} = \frac{1}{2M}\left(p_j - qA_j(\mathbf{x})\right)G^{jk}(\mathbf{x})\left(p_k - qA_k(\mathbf{x})\right) + V\big(\boldsymbol{g}(\mathbf{x})\big), \tag{37}$$

where $G^{jk}(\mathbf{x})$ is the (inverse) metric induced by the deformation $\boldsymbol{g}$:

$$G_{ij}(\mathbf{x}) \equiv \frac{\partial g^k}{\partial x^i}\frac{\partial g^k}{\partial x^j} \qquad \Rightarrow \qquad G^{ij}(\mathbf{x}) = \left(\frac{\partial(g^{-1})^i}{\partial y^k}\frac{\partial(g^{-1})^j}{\partial y^k}\right)_{\mathbf{y}=\boldsymbol{g}(\mathbf{x})}. \tag{38}$$

The operator (37) is thus the deformed Hamiltonian produced by $\boldsymbol{g} \in \mathrm{SDiff}\,\mathbb{R}^2$, again involving a modified metric ($\delta^{ij} \to G^{ij}$) and a deformed potential ($V \to V \circ \boldsymbol{g}$). Note that the vector potential is unchanged in (37), as guaranteed by the compensating gauge transformation in (32)–(35). We stress, in particular, that the transformation (38) in the case of linear maps (2) reduces to the metric redefinitions considered in [19, eq. (10)] and [20, eq. (3.2)]: the corresponding linear response will eventually coincide with the Hall viscosity.

**Aharonov-Bohm extension.** A natural question at this point is whether the assignment (35) furnishes a representation of the group $\mathrm{SDiff}\,\mathbb{R}^2$, since this is a necessary condition for the arguments of sec. 2.1 to hold. To answer this, compose two operators of the form (35) and find that they satisfy eq. (8) with a *non-zero* central extension

$$\frac{\hbar}{q}C(\boldsymbol{g},\boldsymbol{f}) = \int_{\gamma_{\boldsymbol{g}\circ\boldsymbol{f}}}\mathbf{A} - \int_{\boldsymbol{g}(\gamma_{\boldsymbol{f}})}\mathbf{A} - \int_{\gamma_{\boldsymbol{g}}}\mathbf{A}. \tag{39}$$

One may view this as an AB phase, since it is the holonomy of $\mathbf{A}$ along the (curved) triangle $\gamma_{\boldsymbol{g}\circ\boldsymbol{f}} - \boldsymbol{g}(\gamma_{\boldsymbol{f}}) - \gamma_{\boldsymbol{g}}$, *i.e.* the flux of the magnetic field through the surface enclosed by that triangle: see fig. 8. In particular, it is manifestly gauge-invariant. Its presence states that the

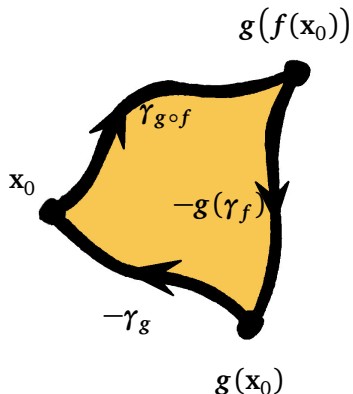

Figure 8: The central extension (39) measures the area of a triangle built out of the paths $\gamma_{g \circ f}$, $-g(\gamma_f)$ and $-\gamma_g$, *i.e.* the flux of the magnetic field through that triangle. This is a special case of a standard geometric construction in group cohomology (see *e.g.* [72, prop. 4.2] or [73] and references therein). In particular, the cocycle identity (9) that ensures associativity is automatically satisfied, since it amounts to the statement that the area of a curved quadrilateral can be decomposed in two different ways as the sum of the areas of two triangles.

operators (35) furnish a *projective* representation of the group of area-preserving maps, implying that their Berry phases have inhomogeneous term as in eq. (10); this will be important in sec. 4.2. Note that the cocycle (39) is non-trivial, *i.e.* it cannot be absorbed by a redefinition of the operators $\mathcal{U}[g]$. One can prove this by noting that (39) reduces to the (obviously non-trivial) Heisenberg central extension of $\mathbb{R}^2$ in the case of magnetic translations (34), as we now confirm by investigating infinitesimal transformations.

**Infinitesimal quantomorphisms.** We will soon need to implement infinitesimal diffeomorphisms (*i.e.* divergence-free vector fields) in quantum mechanics. Accordingly, define Hermitian operators $\mathfrak{u}[v] \equiv -i\partial_\epsilon\big|_0 \mathcal{U}[e^{\epsilon v}]$ as in eqs. (7) and (12), where $v$ is any divergence-free vector field. A direct computation starting from the main definition (35) then yields the differential operator [74, 75]

$$\mathfrak{u}[v] = -i\nabla_v - \frac{q}{\hbar}F_v, \qquad \text{with} \qquad F_v(\mathbf{x}) \equiv \int_{\mathbf{x}_0}^{\mathbf{x}} \iota_v \mathbf{B}. \tag{40}$$

Here the covariant derivative $\nabla_v = v^j(\partial_j - i\frac{q}{\hbar}A_j)$ is a local combination of mechanical momenta $\mathbf{p} - q\mathbf{A}$, generating parallel transport of charged wave functions. As for $F_v$, it is a stream function that originates from the compensating gauge transformation introduced in eq. (32). Note that $F_v$ is now fixed uniquely by the condition $F_v(\mathbf{x}_0) = 0$, so changing the reference point $\mathbf{x}_0$ shifts the stream function $F_v$ by an additive constant. This reflects the unavoidable global phase ambiguity in the definition (35) of $\mathcal{U}[g]$, which in turns allows the representation to be projective.

Let us illustrate eq. (40) with the generators of deformations listed around eq. (30), taking $\mathbf{x}_0 = 0$ for simplicity. As a first example, translations are generated by constant vector fields $v = \partial_i$, with linear stream functions $F_{\partial_i}(\mathbf{x}) = B\varepsilon_{ij}x^j$. The corresponding Hermitian operator (40),

$$\hbar\mathfrak{u}[\partial_j] = p_j - qA_j(\mathbf{x}) - qB\varepsilon_{jk}x^k \qquad \text{(translations)}, \tag{41}$$

is nothing but the gauge-invariant generator of magnetic translations, and could also have been obtained by differentiating eq. (34) with respect to the components of the translation

vector **a**. Note that this implies the usual Heisenberg commutator of magnetic translations, $[\hbar\mathfrak{u}[\partial_x],\hbar\mathfrak{u}[\partial_y]]=-i\hbar qB$, confirming that the cocycle (39) is non-trivial. This is actually a special case of the commutator of arbitrary operators (40),

$$\big[\mathfrak{u}[\boldsymbol{v}],\mathfrak{u}[\boldsymbol{w}]\big]=i\mathfrak{u}\big[[\boldsymbol{v},\boldsymbol{w}]\big]-\tfrac{iq}{\hbar}\mathbf{B}(\boldsymbol{v},\boldsymbol{w})\big|_{\mathbf{x}_0},\tag{42}$$

which exhibits a 'Lichnerowicz' central charge due to the magnetic field (see *e.g.* [76]).

A less elementary example that will be essential below is provided by edge deformations (25) whose vector fields (27) and stream functions (30) are uniquely labelled by a 1D vector field $v(\varphi)\partial_\varphi$. Written in an arbitrary gauge, the corresponding Hermitian operators (40) are

$$\mathfrak{u}[\boldsymbol{v}]=-iv(\varphi)\partial_\varphi+i\frac{r}{2}v'(\varphi)\partial_r-\frac{q}{\hbar}\langle\mathbf{A},\boldsymbol{v}\rangle+\frac{qB}{2\hbar}r^2v(\varphi)\qquad\text{(edge deformations)},\tag{43}$$

where $\langle\mathbf{A},\boldsymbol{v}\rangle\equiv\iota_{\boldsymbol{v}}\mathbf{A}\equiv v^iA_i$ is the pairing of the one-form $\mathbf{A}$ with the vector field $\boldsymbol{v}$, and the stream function (30) is apparent on the right-hand side. This simplifies neatly in symmetric gauge: when $\mathbf{A}=\frac{1}{2}Br^2\mathrm{d}\varphi$, the term $\langle\mathbf{A},\boldsymbol{v}\rangle$ cancels the stream function and eq. (43) becomes

$$\mathfrak{u}[\boldsymbol{v}]=-iv(\varphi)\partial_\varphi+i\frac{r}{2}v'(\varphi)\partial_r\qquad\begin{pmatrix}\text{edge deformations}\\\text{in symmetric gauge}\end{pmatrix}.\tag{44}$$

At the level of the group $\mathrm{SDiff}\,\mathbb{R}^2$, the simplification means that the action (35) of edge deformations in symmetric gauge reduces to the naive ansatz (31). This will greatly simplify the identification of the Hall viscosity in our setup.

# 4 Adiabatic deformations of planar droplets

This section presents our main result: explicit Berry phases due to adiabatic area-preserving deformations acting unitarily on a many-body droplet of electrons in the plane. As a preliminary, we first display such phases for electrically neutral states. Adding a magnetic field produces the extension (39), and the associated one-body phases can be derived following eq. (10). They turn out to consist of two terms, both separately gauge-invariant (see eq. (49)): the first involves the current density, and the second is an Aharonov-Bohm (AB) phase sensitive to the wave function's probability density. Their many-body generalization thus involves the current and density of the full droplet (see eq. (50)).

We eventually apply this to edge deformations (25) and observe that the AB phase is superextensive in the thermodynamic limit ($\propto N^2$ for $N\gg1$ electrons). By contrast, the contribution of the current is *extensive* at very strong magnetic fields, *i.e.* for genuine quantum Hall (QH) droplets. The corresponding finite Berry curvature per unit area measures the jump of the current at the edge and is reminiscent of the Hall viscosity, with which it coincides up to an overall factor for both integer and fractional QH states. The Hall viscosity as such is recovered by restricting attention to linear deformations of the plane that only affect the metric without touching the potential.

## 4.1 Invitation: Berry phases of neutral states

To start, let us compute Berry phases due to deformations of *neutral* planar wave functions. This preliminary will exhibit a key aspect of the charged case as well: geometric phases measure current. As in sec. 3, we only consider area-preserving maps, whose action on wave functions is now given by eq. (31) since $q=0$. This induces deformations of the metric and

the one-body potential, as in eq. (37), albeit without vector potential **A**. The corresponding infinitesimal operators are given by the $q = 0$ version of (40):

$$\mathfrak{u}[\boldsymbol{v}] = \frac{1}{\hbar} v^j p_j = -i v^j \partial_j \,, \tag{45}$$

for any divergence-free vector field $\boldsymbol{v}$. The commutators of these operators reproduce the Lie bracket of vector fields, since eq. (42) holds with $q = 0$; there is no central extension.

**Berry phases from deformations.** Let $\Psi$ be a normalized eigenfunction of some one-body Hamiltonian (36) with $q = 0$, and assume its energy is isolated and non-degenerate; this is typically the case if $\Psi$ has sufficiently low energy and the one-body potential $V(\mathbf{x})$ is bounded from below, with few enough symmetries. Let $\boldsymbol{g}_t$ be a closed curve of area-preserving maps, and assume these deformations act adiabatically through eq. (31) on the system initially prepared in the state $\Psi$. What is the ensuing Berry phase?

The answer is provided by the same derivation as in the 1D case of sec. 2.2, and can similarly be obtained in two equivalent ways. The first is to use the middle formula of eq. (6), write the scalar product as a planar integral, rely on the unitary action (31), and evaluate the time derivative by brute force. The second relies on the Lie-algebraic expression on the far right-hand side of eq. (6), using the infinitesimal operator (45) and replacing $\boldsymbol{v}$ by the time derivative (28) of the transformations $\boldsymbol{g}_t$. Regardless of one's approach, the result is

$$\mathcal{B}_\Psi[\boldsymbol{g}_t] = -\frac{M}{\hbar} \oint dt \int d^2\mathbf{x} \left\langle \boldsymbol{j}, \dot{\boldsymbol{g}}_t \circ \boldsymbol{g}_t^{-1} \right\rangle = -\frac{M}{\hbar} \oint dt \int d^2\mathbf{x} \left\langle \boldsymbol{j}, \boldsymbol{v}_t \right\rangle, \tag{46}$$

where $\boldsymbol{j} = \frac{\hbar}{2Mi}(\Psi^* d\Psi - \Psi d\Psi^*)$ is the probability current of $\Psi$, seen as a one-form so that $\langle \boldsymbol{j}, \boldsymbol{v} \rangle \equiv \iota_{\boldsymbol{v}} \boldsymbol{j} = v^i j_i$ denotes the pairing of a one-form with a vector field, as in (43).

Similarly to the 1D case of sec. 2.2, we view the phase (46) as a functional of the path $\boldsymbol{g}_t$ that depends parametrically on the current $\boldsymbol{j}$. In particular, states that carry no current have vanishing Berry phases. The AB effect will change this conclusion for charged wave functions in a magnetic field.

**Many-body phases.** The generalization of eq. (46) to fermionic or bosonic Fock spaces is immediate. Indeed, suppose $\Psi(\mathbf{x}_1, \mathbf{x}_2, ..., \mathbf{x}_N)$ is the ground state wave function for $N$ particles governed by some many-body Hamiltonian, possibly including interactions. One can then act on $\Psi$ with an $N$-fold tensor product of time-dependent unitary deformations (35) and use the adiabatic theorem [65,77] to obtain the resulting Berry phases just as we did in sec. 2.1. In the case at hand, these phases are given by (46) up to the replacement of the one-body probability current $\boldsymbol{j}$ by the many-body numerical current density $\boldsymbol{J}$:

$$\mathcal{B}[\boldsymbol{g}_t] = -\frac{M}{\hbar} \oint dt \, d^2\mathbf{x} \left\langle \boldsymbol{J}, \dot{\boldsymbol{g}} \circ \boldsymbol{g}^{-1} \right\rangle. \tag{47}$$

One has *e.g.* $\boldsymbol{J} = \sum_{i=1}^N \boldsymbol{j}_i$ for free electrons, where the sum runs over occupied one-body states; but we emphasize that the phase formula (47) holds even when the constituent particles interact. In any case, the absence of current immediately implies $\mathcal{B} = 0$.

## 4.2 Berry phases of charged states

Let us now ask how eqs. (46)–(47) generalize to charged droplets. As in sec. 4.1, we start by deriving Berry phases for one-body wave functions; the many-body generalization will then be straightforward. This time, however, we first exploit the intuition provided by eq. (46) to avoid computations and 'guess' the form of deformational Berry phases for charged states. This is then confirmed by a detailed proof based on the unitary action (35).

**Inferring Berry phases.**  The neutral result (46) exhibits the dependence of Berry phases on current. A similar phenomenon may be anticipated for charged states, except that the current in (46) must be replaced by its gauge-invariant version

$$\boldsymbol{j} = \tfrac{\hbar}{2Mi}\big(\Psi^*\nabla\Psi - (\nabla\Psi)^*\Psi\big) = \tfrac{\hbar}{2Mi}\big(\Psi^*(\mathrm{d}-iq\mathbf{A}/\hbar)\Psi - \Psi(\mathrm{d}+iq\mathbf{A}/\hbar)\Psi^*\big). \tag{48}$$

The integral (46) with a current given by (48) is thus gauge-invariant, but it cannot be the complete Berry phase of a charged wave function acted upon by adiabatic deformations. Indeed, the AB effect [51] yields phases that even affect states carrying no current (think for instance of Gaussian wave functions subjected to time-dependent translations). One therefore expects an additional AB contribution to the earlier phase (46), resulting in

$$\mathcal{B}_\Psi[\boldsymbol{g}_t] = -\frac{M}{\hbar}\oint \mathrm{d}t\,\mathrm{d}^2\mathbf{x}\,\langle \boldsymbol{j}, \dot{\boldsymbol{g}}_t\circ \boldsymbol{g}_t^{-1}\rangle + \frac{q}{\hbar}\int \mathrm{d}^2\mathbf{x}\,|\Psi(\mathbf{x})|^2 \oint_{\boldsymbol{g}_t^{-1}(\mathbf{x})}\mathbf{A}, \tag{49}$$

where the current $\boldsymbol{j}$ in (48) is gauge-invariant, as are the density $|\Psi|^2$ and the holonomy of $\mathbf{A}$ in the second term. This satisfies both the requirement that Berry phases measure current and the existence of the AB effect. The presence of a holonomy along the *inverse path* $\boldsymbol{g}_t^{-1}$ in (49) is a technical detail that stems from the definition (35): had all $\boldsymbol{g}$'s on the right-hand side of that formula been replaced by $\boldsymbol{g}^{-1}$'s, one would have obtained a left group action instead of a right one, and the AB holonomy in (49) would have read $\oint_{\boldsymbol{g}_t(\mathbf{x})}\mathbf{A}$.

As before, the many-body generalization of (49) is straightforward since the action of deformations on $N$-body wave functions is an $N$-fold tensor product of one-particle formulas (35). The current $\boldsymbol{j}$ in (49) is thus replaced by its many-body analogue $\boldsymbol{J}$ and the probability density $|\Psi|^2$ is replaced by the many-body numerical density $\rho(\mathbf{x})$, resulting in the many-body Berry phase

$$\boxed{\mathcal{B}[\boldsymbol{g}_t] = -\frac{M}{\hbar}\oint \mathrm{d}t\,\mathrm{d}^2\mathbf{x}\,\langle \boldsymbol{J}, \boldsymbol{v}_t\rangle + \frac{q}{\hbar}\int \mathrm{d}^2\mathbf{x}\,\rho(\mathbf{x})\oint_{\boldsymbol{g}_t^{-1}(\mathbf{x})}\mathbf{A}.} \tag{50}$$

For free electrons, $\rho(\mathbf{x}) = \sum_{i=1}^N |\Psi_i(\mathbf{x})|^2$ is a sum over occupied one-body states, but we stress again that eq. (50) holds even for interacting states. The key point is that (50) is only sensitive to two universal properties of any charged droplet: its current and density.

**Deriving Berry phases.**  Having guessed eq. (49), let us prove that it follows (in the sense of sec. 2.1) from the action (35) of deformations on wave functions. The computation highlights the geometric structure of quantomorphisms, but does not affect later applications; the hasty reader may therefore skip it and go straight to sec. 4.3.

We have seen that eq. (35) furnishes a *projective* representation of area-preserving maps, so the ensuing Berry phase is given by (10) and involves a term due to the extension (39):

$$\mathcal{B}_\Psi[\boldsymbol{g}_t] = \oint \mathrm{d}t\,\langle\Psi|\big(i\nabla_{\boldsymbol{v}_t} + \tfrac{q}{\hbar}F_{\boldsymbol{v}_t}\big)|\Psi\rangle - \oint \mathrm{d}t\,\partial_\tau \mathsf{C}(\boldsymbol{g}_\tau, \boldsymbol{g}_t^{-1})\big|_{\tau=t}, \tag{51}$$

where $\boldsymbol{v}_t$ is the velocity vector field (28) and we used the expression (40) of the Lie algebra operator $\mathfrak{u}[\boldsymbol{v}_t]$. The covariant derivative then gives rise to the gauge-invariant current (48), which yields the first piece of the phase on the right-hand side of (49):

$$\mathcal{B}_\Psi[\boldsymbol{g}_t] = -\frac{M}{\hbar}\oint \mathrm{d}t\,\mathrm{d}^2\mathbf{x}\,\langle \boldsymbol{j}, \boldsymbol{v}_t\rangle + \frac{q}{\hbar}\oint \mathrm{d}t\,\mathrm{d}^2\mathbf{x}\,|\Psi(\mathbf{x})|^2 F_{\boldsymbol{v}_t}(\mathbf{x}) - \oint \mathrm{d}t\,\partial_\tau \mathsf{C}(\boldsymbol{g}_\tau, \boldsymbol{g}_t^{-1})\big|_{\tau=t}. \tag{52}$$

It only remains to evaluate the contribution of the stream function and that of the central extension. We begin with the former and rely on its definition in (40) to write

$$\oint dt\, F_{v_t}(\mathbf{x}) = \oint dt \int_{\mathbf{x}_0}^{\mathbf{x}} \iota_{\dot{g}_t \circ g_t^{-1}} \mathbf{B} = \oint dt \int_{\mathbf{x}_0}^{\mathbf{x}} \iota_{\dot{g}_t \circ g_t^{-1}} \big( (g_t^{-1})^* \mathbf{B} \big), \tag{53}$$

where the second equality was obtained thanks to the fact that $g_t$'s preserve area. Since the integral from $\mathbf{x}_0$ to $\mathbf{x}$ is taken along any path connecting them, let $\gamma(s)$ be such a path with $\gamma(0) = \mathbf{x}_0$ and $\gamma(1) = \mathbf{x}$, and unpack the integral on the far right-hand side of (53) as

$$\oint dt\, F_{v_t}(\mathbf{x}) = -\oint_0^T dt \int_0^1 ds\, \mathbf{B}_{g_t^{-1}(\gamma(s))} \Big( \frac{\partial}{\partial t} g_t^{-1}(\gamma(s)), \frac{\partial}{\partial s} g_t^{-1}(\gamma(s)) \Big). \tag{54}$$

This is a surface integral of the magnetic field $\mathbf{B}$, with two boundaries (see fig. 9): one is the loop $g_t^{-1}(\mathbf{x})$, the other is $g_t^{-1}(\mathbf{x}_0)$. By Stokes's theorem, eq. (54) can thus be recast as

$$\oint dt\, F_{v_t}(\mathbf{x}) = \oint_{g_t^{-1}(\mathbf{x})} \mathbf{A} - \oint_{g_t^{-1}(\mathbf{x}_0)} \mathbf{A}, \tag{55}$$

which is nearly what we need to match eq. (52) with the expected result (49): the only difference between the two now resides in the holonomy of $\mathbf{A}$ along $g_t^{-1}(\mathbf{x}_0)$ in the very last term of eq. (55). This is where the cocycle (39) finally comes to the rescue: we saw in fig. 8 that it measures the area of a (curved) triangle. In the case at hand, eq. (52) involves

$$\frac{\hbar}{q} \mathsf{C}(g_\tau, g_t^{-1}) = \quad \begin{array}{c} g_\tau(g_t^{-1}(\mathbf{x}_0)) \\[-2pt] \end{array} \quad g_\tau(\mathbf{x}_0) \ = \quad \begin{array}{c} g_t^{-1}(\mathbf{x}_0) \\[-2pt] \end{array} \quad \mathbf{x}_0 \tag{56}$$

where the second equality was obtained by acting on the left triangle with the area-preserving map $g_\tau^{-1}$. One can then differentiate $\mathsf{C}(g_\tau, g_t^{-1})$ to find $\partial_\tau \mathsf{C}(g_\tau, g_t^{-1}) = -\frac{q}{\hbar} \mathbf{A}_{g_t^{-1}(\mathbf{x}_0)} (\partial_t g_t^{-1}(\mathbf{x}_0))$ up to a total time derivative, since $\tau$ only appears in the argument of the vertex at $g_\tau^{-1}(\mathbf{x}_0)$ (see app. B for details). Integrating over $t$ then cancels the holonomy along $g_t^{-1}(\mathbf{x}_0)$ in (55), eventually reproducing the AB term in eq. (49).

## 4.3 Examples of adiabatic deformations

Eq. (50) is general and holds for any sequence of deformations, so we can apply it to specific families of area-preserving maps (recall the examples of sec. 3.1). We first study translations and rotations, then turn to edge deformations (25).

**Translations and rotations.**   Consider a sequence of time-dependent translations $\mathbf{x} \mapsto \mathbf{x} + \mathbf{a}_t$ (with $\mathbf{a}_T = \mathbf{a}_0$). Their only effect on the Hamiltonian (37) is to shift the location of the potential, since the kinetic term is translation-invariant. The same can be confirmed in terms of Berry phases: the velocity vector field (28) of overall translations is a total derivative $v_t = \dot{\mathbf{a}}_t$, so the current integral in the Berry phase (50) vanishes and only the AB contribution remains. The latter actually involves the same holonomy $\oint_{\mathbf{a}_t} \mathbf{A}$ at all positions (since translations do not depend on $\mathbf{x}$), and one eventually finds

$$\mathcal{B}[g_t] = -\frac{q}{\hbar} N \oint_{\mathbf{a}_t} \mathbf{A} \qquad \text{(translations)}, \tag{57}$$

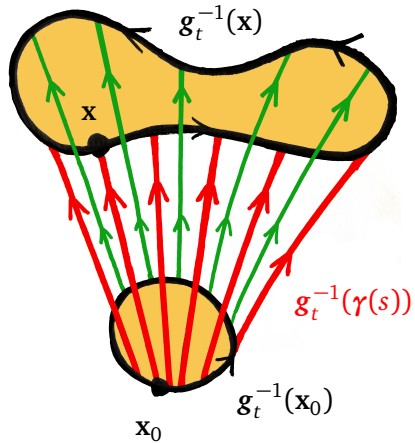

Figure 9: In (54), the magnetic field $\mathbf{B} = \mathrm{d}\mathbf{A}$ is integrated over a surface whose points are $\boldsymbol{g}_t^{-1}(\boldsymbol{\gamma}(s))$, where $t \in [0, T]$, $s \in [0, 1]$ and $\boldsymbol{\gamma}(s)$ connects some origin $\mathbf{x}_0$ to the point $\mathbf{x}$. Each $\boldsymbol{g}_t$ is a deformation and $\boldsymbol{g}_T = \boldsymbol{g}_0$; for definiteness, the cartoon on the left assumes $\boldsymbol{g}_0 = e$ the identity, but this is not required in the proof of eq. (50). The time parameter $t$ increases along the black curves, so the two black loops are the images of $\boldsymbol{g}_t^{-1}(\mathbf{x}_0)$ and $\boldsymbol{g}_t^{-1}(\mathbf{x})$ for $t \in [0, T]$. The auxiliary parameter $s$ increases along red and green lines, which are thus paths of the form $\boldsymbol{g}_t^{-1}(\boldsymbol{\gamma}(s))$ at fixed $t$. The simplification of (54) in two holonomies (55) follows from the cancellation between integrals over red and green curves when these curves overlap, which only leaves out the two yellow surfaces with boundaries $\boldsymbol{g}_t^{-1}(\mathbf{x}_0)$ and $\boldsymbol{g}_t^{-1}(\mathbf{x})$.

where $N = \int \mathrm{d}^2\mathbf{x}\, \rho(\mathbf{x})$ is the total number of particles in the droplet. This is manifestly just an overall AB phase, as was to be expected (with a minus sign stemming from our choice of right group actions in eqs. (3) and (35)). One can think of it as the response of a charged droplet to adiabatic changes of the location of the potential well.

Another elementary example is provided by rotations around the origin: in that case, both the current and the density generally contribute to the phase (50). Using polar coordinates $(r, \varphi)$, adiabatic rotations read $\varphi \mapsto \varphi + \theta_t$ with $\theta_T = \theta_0 + 2\pi n$ for some integer $n$. The AB holonomy of (50) thus reads $\oint \mathbf{A} = -Br^2\pi n$ and the velocity vector field (28) is purely angular: $\dot{\boldsymbol{g}}_t \circ \boldsymbol{g}_t^{-1} = \dot{\theta}_t\, \partial_\varphi$. Thus the current one-form

$$\boldsymbol{J} = J_\varphi(r, \varphi)\mathrm{d}\varphi + J_r(r, \varphi)\mathrm{d}r\,, \tag{58}$$

only contributes to the phase (50) through its angular component $J_\varphi$, and the complete phase (50) for adiabatic rotations reads

$$\mathcal{B}[\boldsymbol{g}_t] = -2\pi n \frac{M}{\hbar} \int \mathrm{d}^2\mathbf{x}\Big[J_\varphi(r, \varphi) + \frac{\omega_c}{2}r^2\rho(r, \varphi)\Big] \qquad \text{(rotations)}, \tag{59}$$

where $\omega_c \equiv qB/M$ is the cyclotron frequency. This generalizes the 1D result (20) and similarly measures the average current, save for an additional density contribution that now makes the Berry phase super-extensive in the thermodynamic limit. Note that (59) *vanishes* modulo $2\pi$ in the case of isotropic states. Indeed, for one-body wave functions, its integrand

$$J_\varphi + \frac{\omega_c}{2}r^2\rho = \frac{\hbar}{2Mi}(\Psi^*\partial_\varphi\Psi - \Psi\partial_\varphi\Psi^*), \tag{60}$$

is just $\hbar/M$ times the angular momentum of $\Psi$. Since angular momentum is an integer, the phase (59) is indeed an integer multiple of $2\pi$. The same is true of isotropic states whose

total angular momentum is the sum of angular momenta of their constituents. Following the remark below eq. (7), this means that the parameter space for deformations of isotropic states is (at most) a quotient space $\mathrm{SDiff}\,\mathbb{R}^2/S^1$.

**Edge deformations.** The maps (25) are 'rotations with extra wiggles', so their Berry phases are expected to generalize eq. (59). Indeed, working once more in symmetric gauge $\mathbf{A} = \frac{1}{2}Br^2\mathrm{d}\varphi$, the AB holonomy of eq. (50) is

$$\oint_{\boldsymbol{g}_t^{-1}(\mathbf{x})} \mathbf{A} = -\frac{Br^2}{2}\oint \mathrm{d}t\,\dot{g}_t\big(g_t^{-1}(\varphi)\big), \tag{61}$$

where the $g_t(\varphi)$'s are 1D diffeomorphisms that determine adiabatic edge deformations (25). The appearance of a 1D Berry phase analogous to eq. (16) is thus manifest. As for the fluid velocity (28) of edge deformations, it is a vector field of the form (27):

$$\dot{\boldsymbol{g}}_t \circ \boldsymbol{g}_t^{-1} \equiv \boldsymbol{v}_t = \dot{g}\big(g^{-1}(\varphi)\big)\partial_\varphi - \frac{r}{2}\big(\dot{g}\circ g^{-1}\big)'(\varphi)\partial_r, \qquad \text{(edge deformations).} \tag{62}$$

It only remains to pair this with the current (58), yielding the Berry phase

$$\mathcal{B}[\boldsymbol{g}_t] = -\frac{M}{\hbar}\oint \mathrm{d}t\,\mathrm{d}\varphi \int_0^\infty r\,\mathrm{d}r\bigg[J_\varphi(r,\varphi) + \frac{r}{2}\partial_\varphi J_r(r,\varphi) + \frac{\omega_c}{2}r^2\rho(r,\varphi)\bigg]\dot{g}_t\big(g_t^{-1}(\varphi)\big). \tag{63}$$

Remarkably, this is just a phase (16) produced by 1D deformations $g_t(\varphi)$, now involving an effective 1D current

$$J_{\mathrm{eff}}(\varphi) \equiv \int_0^\infty r\,\mathrm{d}r\bigg[J_\varphi(r,\varphi) + \frac{r}{2}\partial_\varphi J_r(r,\varphi) + \frac{\omega_c}{2}r^2\rho(r,\varphi)\bigg]. \tag{64}$$

As in eq. (59), the contribution involving $r^2 \times \rho$ in (64) is super-extensive in the thermodynamic limit. Furthermore, a simplification occurs again in isotropic states, for which (64) is $\varphi$-independent and reduces to $\hbar/M$ times angular momentum.

## 4.4 Deformations of Hall droplets and Hall viscosity

Eq. (50) holds for any charged many-body ground state in the plane, so we now apply it to QH droplets with finite area consisting of $N$ spin-polarized electrons. We begin by recalling basic aspects of planar quantum mechanics in the limit of strong magnetic fields, then study the Berry phases (50) due to edge deformations in the same regime. This leads to the observation that the current contribution to the Berry phase is extensive in the thermodynamic limit. From there we derive an analogue of the Hall viscosity thanks to the effective current (64), and eventually show how Hall viscosity itself is recovered in our formalism.

**Landau levels and QH droplets.** Consider the Hamiltonian (36) with a strong magnetic field $\mathbf{B} = \mathrm{d}\mathbf{A}$ and a weak confining potential $V(\mathbf{x})$. For simplicity, we assume that $V(\mathbf{x})$ is isotropic and harmonic, so $V(\mathbf{x}) = \kappa r^2/2$ for some 'stiffness' $\kappa \ll M\omega_c^2$.[12] Then the energy spectrum can be organized in well-defined Landau levels, with non-degenerate energies thanks to the presence of $V(\mathbf{x})$. Indeed, in symmetric gauge $\mathbf{A} = \frac{1}{2}Br^2\mathrm{d}\varphi$, an orthonormal basis of energy eigenstates with angular momentum $\hbar(m-n)$ in the $n^{\mathrm{th}}$ level is given by wave functions that involve generalized Laguerre polynomials $L_n^a(x) \equiv \frac{1}{n!}x^{-a}(\partial_x - 1)^n x^{n+a}$,

$$\Psi_{n,m}(\mathbf{x}) = \frac{1}{\sqrt{2\pi\ell^2}}\sqrt{\frac{n!}{m!}}\,z^{m-n}L_n^{m-n}(|z|^2)\,e^{-|z|^2/2}. \tag{65}$$

---

[12]We will eventually assume the stronger relation $\kappa \ll M\omega_c^2/N$ for $N \gg 1$ electrons: see eqs. (68)–(69).

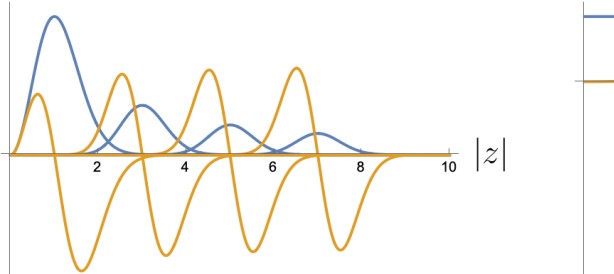 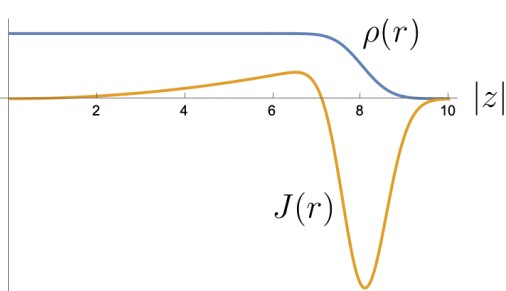

Figure 10: *Left:* Probability densities (blue) and current densities (yellow) of states in the lowest Landau level, given by eqs. (65)–(66) with $n = 0$ and $m = 1, 9, 25, 49$. *Right:* Many-body density (blue) and many-body current (yellow) of isotropic droplets, obtained by summing over the densities and currents of states (65) with $n = 0$ and $m$ ranging from 0 to $N - 1 = 64$. The density is constant for $|z| \lesssim \sqrt{N}$ and drops to zero at the edge $|z| \sim \sqrt{N}$, while the current grows slowly following the gradient of the potential in the bulk, then 'jumps' near the edge. Similar plots hold upon including higher Landau levels, save for additional oscillations in both one-body and many-body observables (see *e.g.* [79]).

Here $\ell^2 \equiv \hbar / \sqrt{q^2 B^2 + 4M\kappa}$ is the (squared) magnetic length corrected by harmonic effects and $z \equiv (x + iy)/\sqrt{2\ell^2}$ is a dimensionless complex coordinate. At very strong magnetic fields, the energy of each such state is $\hbar \omega_c n + \kappa \ell_B^2 m$ up to an irrelevant additive constant, where $\ell_B \equiv \sqrt{\hbar/qB}$ is the standard magnetic length. The corresponding (gauge-invariant) probability current (48) is purely angular:

$$ \boldsymbol{j}_{n,m} = \frac{\hbar}{M} \frac{1}{2\pi\ell^2} \frac{|z|^{2m-2n}}{m!} n! \left[ L_n^{m-n}(|z|^2) \right]^2 e^{-|z|^2} \left( m - n - \frac{\ell^2}{\ell_B^2} |z|^2 \right) \mathrm{d}\varphi \,. \tag{66} $$

Both the density $|\Psi_{n,m}(\mathbf{x})|^2$ and the current only depend on the radius $|z|$, and they are localized around $|z| = \sqrt{m}$ (see the left panel of fig. 10 for the lowest level, $n = 0$). Qualitatively similar conclusions hold in weak *anharmonic* but isotropic traps [78].

Now let an isotropic QH droplet consist of $N \gg 1$ occupied one-body states (65) with the lowest possible energy. These states belong to some subset $\{0, 1, ..., \nu - 1\}$ of the available Landau levels, where $\nu$ is some integer filling fraction. In the thermodynamic limit, each level contains $\sim N/\nu$ occupied states. Then the many-body density $\rho(r)$ is nearly constant and equal to $\nu/(2\pi\ell^2)$ in the bulk, before falling sharply to zero near the edge located at $r_{\text{edge}} = \sqrt{2\ell^2 N/\nu}$ (see the right panel of fig. 10). The ground state thus forms a disk-shaped droplet with roughly uniform density and area $2\pi\ell^2 N/\nu$. As for the purely angular current $\boldsymbol{J} = J_\varphi(r)\mathrm{d}\varphi$, its bulk behaviour is determined by the Hall law $J_\varphi(r) = \frac{\nu}{2\pi\hbar} r \partial V/\partial r$ in terms of the filling fraction $\nu$, followed by a robust jump localized at the edge [52, 53].

**Adiabatic edge deformations.** Knowing currents and densities, the Berry phases (50) follow for any sequence of adiabatic deformations. For instance, bulk effects are exhibited by deformations that only affect the interior of the droplet while fixing its boundary, as in fig. 7 above. Conversely, the jump of the current at the edge contributes to Berry phases produced by deformations $\boldsymbol{g}_t$ that move the boundary without affecting the bulk. Let us now focus on edge deformations (25), whose Berry phases are given in full generality by eq. (63). As we saw around eq. (60), the resulting 1D current (64) is really just the total angular momentum (since the droplet is isotropic). The one-body phase (63) for a state (65) thus reads

$\mathcal{B}_{n,m}[\mathbf{g}_t] = -(m-n)\oint \frac{dt\, d\varphi}{2\pi}\, \dot{g}_t\big(g_t^{-1}(\varphi)\big)$, and its many-body generalization is

$$\mathcal{B}[\mathbf{g}_t] = -L \oint \frac{dt\, d\varphi}{2\pi}\, \dot{g}_t\big(g_t^{-1}(\varphi)\big), \qquad \text{with} \qquad L \sim \frac{N^2}{2\nu}. \tag{67}$$

Here $L$ is the droplet's total angular momentum when the occupied Landau levels are $n = 0, 1, ..., \nu-1$, each containing $\sim N/\nu$ states in the thermodynamic limit. The Berry phase is thus super-extensive, as observed below eq. (64). Note that this also holds for fractional QH states; for example, the angular momentum of the Laughlin wave function [80] at filling $\nu$ is $L = N(N-1)/(2\nu)$, in accordance with (67).

A weakness of (67) is to completely miss the fact that the total Berry phase (50) contains two pieces: a trivial AB phase, and an additional current term. It is instructive to compute the current contribution separately, which we do now. Accordingly, consider the 'truncated' phase (46), with a gauge-invariant current (48). Applying this to a one-body state (65) yields a phase (67) whose coefficient $L$ is replaced by the integrated current

$$\frac{2\pi M}{\hbar} J_{\text{int}} \equiv \frac{2\pi M}{\hbar} \int_0^\infty r\, dr\, j_{n,m}(r) \overset{(66)}{=} \big(1 - \tfrac{\ell^2}{\ell_B^2}\big)m - \big(1 + \tfrac{\ell^2}{\ell_B^2}\big)n - \tfrac{\ell^2}{\ell_B^2}. \tag{68}$$

This still depends on $m$, so the corresponding sum over $N \gg 1$ occupied states is super-extensive. However, the $m$ dependence is only due to the confining potential and disappears in the limit of strong magnetic fields. Assuming indeed that $\kappa \ll M\omega_c^2/N$, one may expand $\ell^2/\ell_B^2 \sim 1 - 2M\kappa/q^2 B^2$ and find $J_{\text{int}} = -\frac{\hbar}{2\pi M}(2n+1)$, which now only depends on the Landau level $n$ [53, eq. (18)]. The part of the many-body phase (67) due only to the current then becomes extensive:

$$\mathcal{B}_{\text{current}}[\mathbf{g}_t] \sim \sum_{n=0}^{\nu-1}(2n+1) \sum_{m=0}^{N/\nu} \oint \frac{dt\, d\varphi}{2\pi}\, \dot{g}_t\big(g_t^{-1}(\varphi)\big) \sim N\nu \oint \frac{dt\, d\varphi}{2\pi}\, \dot{g}_t\big(g_t^{-1}(\varphi)\big). \tag{69}$$

We stress that the final coefficient $N\nu$ appearing here is really an integral (64) of the (angular component of the) many-body current:

$$\frac{N\nu}{2\pi} = -\frac{M}{\hbar} \int_0^\infty r\, dr\, J_\varphi(r)\Big|_{\nu \text{ levels with } N/\nu \gg 1 \text{ states, infinite } \mathbf{B}}. \tag{70}$$

Eq. (69) is thus an extensive Berry phase that is *independent* of the confining potential, and that stems entirely from the many-body current *at the edge*. In fact, the same contribution would have been obtained for *any* deformation of the droplet that acts non-trivially on its boundary, regardless of the detailed form of edge deformations (25). The coefficient (70) is universal in that sense; we now show that it is analogous to the Hall viscosity.

**Hall viscosity revisited.** The Berry phases (67) and (69) can obviously be applied to explicit edge deformations, as done in sec. 2.3. In particular, the Berry curvature (24) holds for $SL(2,\mathbb{R})$ transformations (21), except that the one-body parameter $s$ is now replaced by the angular momentum $m - n$ in (67), or by the coefficient $-2n - 1$ stemming from the strong magnetic field limit of eq. (68). The Berry curvature for a single state (65) thus reads

$$\mathcal{F}_{n,m} = \tfrac{4}{k}\sinh(2\lambda)\, d\lambda \wedge d\theta \times \begin{cases} n - m\,, & \text{for full phase (67)}, \\ 2n + 1\,, & \text{for pure current phase (69)}. \end{cases} \tag{71}$$

For $k = 2$, this is the curvature associated with *linear* deformations (2) whose Jacobian matrix is constant in Cartesian coordinates. It is then customary [19, 20] to rewrite (71) in terms

of a complex parameter $\tau = \tau_1 + i\tau_2$ (with $\tau_2 > 0$). The latter can be read off from the transformation of the Euclidean metric under linear maps (2), namely [19, eq. (10)]

$$(a\,dx + b\,dy)^2 + (c\,dx + d\,dy)^2 \equiv \frac{1}{\tau_2}\big(dx^2 + 2\tau_1\,dx\,dy + |\tau|^2\,dy^2\big), \tag{72}$$

which yields $\tau = (ab + cd + i)/(a^2 + c^2)$. The correspondence (26) between $(a, b, c, d)$ and the complex parameters $(\alpha, \beta) = (e^{i(\chi+\theta)}\cosh\lambda, e^{i(\chi-\theta)}\sinh\lambda)$ then gives

$$\tau = \frac{-\sin(2\theta)\sinh(2\lambda) + i}{\cosh(2\lambda) + \cos(2\theta)\sinh(2\lambda)}, \tag{73}$$

finally allowing us to rewrite the one-body Berry curvature (71) with $k = 2$ as

$$\mathcal{F}_{n,m} = -\frac{1}{2}\frac{d\tau_1 \wedge d\tau_2}{\tau_2^2} \times \begin{cases} n - m, & \text{for full phase (67),} \\ 2n + 1, & \text{for pure current phase (69).} \end{cases} \tag{74}$$

Note that the 'pure current' result with its coefficient $2n + 1 = 2(n + 1/2)$ reproduces [20, eq. (3.21)] up to an overall factor 2. It is thus very close, but not quite identical, to the Berry curvature normally associated with the Hall viscosity of a state in the $n^{\text{th}}$ Landau level.

This mild discrepancy is not a mere matter of conventions. To see its origin, suppose we had limited the discussion to *linear* deformations (2) from the outset. We would then have found that their unitary action (35) on wave functions reduces to eq. (31) in symmetric gauge, since the latter is preserved by edge deformations (25). The corresponding Lie algebra operators (44) can be expressed as quadratic combinations of ladder operators $a^\dagger = \bar{z}/2 - \partial$ and $b = \bar{z}/2 + \partial$, which respectively raise the Landau level and decrease angular momentum within a level. Indeed, in terms of the coordinates $(\lambda, \theta, \chi)$ defined above eq. (23) and expanding the group element (21) at $k = 2$ up to first order in $\lambda$ and $\chi + \theta$, one has

$$\begin{aligned} \mathfrak{u}[\nu] = &-i\lambda e^{-i\chi+i\theta}a^2 + i\lambda e^{i\chi-i\theta}(a^\dagger)^2 - (\chi + \theta)(a^\dagger a + aa^\dagger) \\ &-i\lambda e^{i\chi-i\theta}b^2 + i\lambda e^{-i\chi+i\theta}(b^\dagger)^2 + (\chi + \theta)(b^\dagger b + bb^\dagger), \end{aligned} \tag{75}$$

for infinitesimal linear deformations. The factorization between $a$ and $b$ pieces is manifest, respectively corresponding to deformations of the metric and the (slowly varying) confining potential in the Hamiltonian (36). The resulting Berry phase (49) consists of two parts, respectively involving expectation values of $\frac{1}{2}(a^\dagger a + aa^\dagger)$ and $\frac{1}{2}(b^\dagger b + bb^\dagger)$. In fact, these two parts can be read off from the first line in the curvature (74) upon writing $n - m = (n + 1/2) - (m + 1/2)$:

$$\mathcal{F}_{n,m,\,\text{full}} = \underbrace{-(n + 1/2)\frac{1}{2}\frac{d\tau_1 \wedge d\tau_2}{\tau_2^2}}_{\mathcal{F}_a} + \underbrace{(m + 1/2)\frac{1}{2}\frac{d\tau_1 \wedge d\tau_2}{\tau_2^2}}_{\mathcal{F}_b}. \tag{76}$$

Here the term involving $n + 1/2$ coincides precisely with [20, eq. (3.21)]; there is no factor 2 mismatch, and it originates indeed from metric deformations produced by the $a$ factors in (75), as in [20, eqs. (3.4)–(3.9)]. Standard formulas for the Hall viscosity then follow in the case of a droplet with $\nu$ filled levels: the sum over occupied states is the same as in (69) and yields an extensive total Berry curvature $\mathcal{F}_a = -\frac{1}{4}N\nu\,d\tau_1 \wedge d\tau_2/\tau_2^2$. In the regime of linear response where $\tau$ is close to $i$, the coefficient of the area form $d\tau_1 \wedge d\tau_2/\tau_2^2$ is interpreted as an odd viscosity $\eta_H$ times the droplet's area $2\pi\ell_B^2 N/\nu$, with the standard value

$$\eta_H = \frac{\hbar}{4}N\nu \times \frac{\nu}{2\pi\ell_B^2 N} = \frac{\hbar}{4}\frac{\nu^2}{2\pi\ell_B^2} \qquad \text{(integer } \nu), \tag{77}$$

for integer QH states [19, 20]. We stress that this was obtained without ever referring to a torus. In particular, the complex number (73) may label any point on the hyperbolic upper half-plane, and should not be interpreted as a modular parameter [81].

At this point, it is natural to wonder if the separation between $a$ and $b$ pieces in (70) has anything to do with the distinct contributions of current and density in the complete Berry phase (50). Indeed, we saw in eq. (74) that the Berry curvature due only to the current coincides with $\mathcal{F}_a$ in (76) up to an overall factor 2. This stems from the integrated current (68): at strong magnetic fields, the coefficient $n - m$ in the full curvature (74) splits as

$$n - m = \underbrace{(2n+1)}_{\text{current}} - \underbrace{(m+n+1)}_{\text{AB phase}}, \tag{78}$$

which is crucially *not* the same splitting as in (76). In particular, the fact that the AB phase itself depends on the level $n$ ultimatly gives rise to a current contribution whose Berry curvature is *twice* the curvature of the Hall viscosity for integer QH states. One should not be concerned about this discrepancy: there is no inherent reason for the current portion of the Berry phase (50) to be related to viscosity. The fact that the corresponding Berry curvatures match up to normalization is simply a result of the unique invariant area form on a hyperbolic plane (recall footnote 1). Moreover, it is important to note that the distinction between $a$ and $b$ contributions in the operator (75) is specific to linear deformations (2), and does *not* align with the separation of (50) into current and AB contributions. Identifying the part of the Berry phase (50) arising solely from metric deformations would pose a significant challenge, extending beyond the scope of the quantomorphisms studied here.

**Fractional QH states.** It is tempting to speculate that this factor 2 is a robust property of QH droplets. Indeed, we have just shown that it holds for integer QH states at strong magnetic fields, allowing us to view the Hall viscosity (77) as a quantized integral

$$\eta_H = - \lim_{\substack{\nu \text{ levels,} \\ N \to \infty, \, \mathbf{B} \to \infty}} \frac{\nu}{2\pi \ell_B^2 N} \times \frac{\pi M}{2} \int_0^\infty r \, dr \, J_\varphi(r) \qquad \text{(integer } \nu\text{)}, \tag{79}$$

that measures the net jump of the current's 'stream function' across the edge of a droplet. A similar integral can be devised for fractional QH states, up to a key change in normalization that spoils the aforementioned factor 2. Let us exhibit this in the case of a Laughlin wave function for $N$ electrons at filling $\nu$: then the full Berry phase for edge deformations is given by (67) with a total angular momentum $L = N(N-1)/(2\nu)$. The latter may be seen as an integral of current and density through the 'effective current' (64), namely

$$\frac{N(N-1)}{2\nu} = \frac{M}{\hbar} \int d^2\mathbf{x} \, J_\varphi(r) + \frac{1}{\ell_B^2} \int d^2\mathbf{x} \, r^2 \rho(r), \tag{80}$$

where nothing depends on $\varphi$ since the Laughlin wave function is isotropic. Here the current integral is the coefficient of the 'pure current' Berry phase meant to mimic the Hall viscosity, and its value can be obtained by plugging in eq. (80) the sum rule [82, eq. (B.2)]

$$\int d^2\mathbf{x} \, r^2 \rho(r) = \ell_B^2 N \frac{N-1+2\nu}{\nu}, \tag{81}$$

for the density of an isotropic Coulomb gas. Much more generally, the current integral is readily found in second quantization for *any* isotropic QH droplet:

$$\int_0^\infty r \, dr \, J_\varphi(r) = \sum_{m=0}^\infty \langle a_m^\dagger a_m \rangle \int_0^\infty r \, dr \, j_{0,m}(r) \overset{(68)}{\sim} -\frac{\hbar}{2\pi M} \sum_{m=0}^\infty \langle a_m^\dagger a_m \rangle = -\frac{\hbar N}{2\pi M}, \tag{82}$$

where $a_m^\dagger$ is the Fock space creation operator for the $m^{\text{th}}$ isotropic orbital in the lowest Landau level, and $\langle ... \rangle$ denotes expectation values in the droplet's ground state. One thus concludes that the 'pure current' Berry curvature associated with linear quantomorphisms of any isotropic QH droplet is

$$\mathcal{F}_{\text{current}} = -\frac{N}{2} \frac{d\tau_1 \wedge d\tau_2}{\tau_2^2}, \tag{83}$$

where we use the same hyperbolic coordinates as in eq. (74). This crucially *differs* from the Berry curvature associated with metric deformations of Laughlin states on a torus, calculated in [23, 27] thanks to the plasma analogy and given by

$$\mathcal{F}_{\text{metric}} = -\frac{N}{4\nu} \frac{d\tau_1 \wedge d\tau_2}{\tau_2^2}, \quad \text{with viscosity} \quad \eta_H = \frac{\hbar}{4} \frac{1}{2\pi\ell_B^2} \qquad (\text{integer } 1/\nu). \tag{84}$$

Relating this value of viscosity to an integral of the current requires that the right-hand side of (79) be divided by $\nu$; eq. (79) is thus *not* valid for fractional QH states, as announced. Put differently, the coefficient of the pure current curvature (83) divided by twice the droplet's area predicts a Hall viscosity $\hbar\nu/(8\pi\ell_B^2)$, which differs from (84) by a factor $\nu$.[13] A similar mismatch occurs, for instance, in the Moore-Read state at filling $\nu = 1/2$, where [27] predicts a Hall viscosity $\eta_H = \frac{3}{2}\hbar/(8\pi\ell_B^2)$ while our universal current Berry curvature (83) predicts a response coefficient $\frac{1}{2}\hbar/(8\pi\ell_B^2)$. As previously mentioned, this mismatch is not contradictory; instead, it is a distinguishing characteristic. Hall viscosity is a specific response to pure metric deformations, whereas our current Berry curvature (83) encompasses both metric and potential perturbations, and fundamentally does not have any direct connection to viscosity.

# 5 Conclusion and outlook

This work was devoted to the Berry phases produced by adiabatic deformations of many-body quantum systems. Specifically, we implemented deformations through unitary operators in the Hilbert space, allowing us to interpret the resulting changes of the Hamiltonian (4) as actual deformations of the metric and potential (recall eqs. (14) and (37)).

In all cases, the response involves the current of the state being acted upon. We applied this to 1D quantum wires with potentially observable consequences, reproducing in an elementary way the Berry phases (16) studied in [50] in conformal field theory. But a much more prominent application of our approach was that of 2D planar droplets in a strong magnetic field. The formalism developed for that situation in sec. 3 was based on unitary area-preserving maps generalizing magnetic translations, known as 'quantomorphisms' in geometric quantization [44, 45]. In particular, we introduced the 'edge deformations' (25) that generalize linear maps (2) and deform the boundary of thermodynamically large isotropic droplets. We then turned to our main result, namely an analytical derivation of explicit Berry phases (50) produced by adiabatic quantomorphisms. As emphasized there, such phases consist of two separately gauge-invariant terms—a current piece and an Aharonov-Bohm (AB) piece. This is true for *any* many-body ground state, regardless of the presence of interactions, and thus applies to both integer and fractional quantum Hall (QH) states.

While our formulas hold for *all* area-preserving deformations, we focused repeatedly on edge deformations, eventually showing that their Berry phases include, besides trivial AB phases, a contribution reminiscent of the Hall viscosity [19, 20]. The latter actually emerges as a subleading contribution to the phase, measuring the response to linear deformations produced by mechanical momenta that change the metric, as opposed to magnetic translations

---

[13]Amusingly, the incorrect value $\hbar\nu/(8\pi\ell_B^2)$ is the one that was initially stated in [22], before the correction to $\eta_H = \hbar/(8\pi\ell_B^2)$ in [23]. It is unclear if this is just a coincidence or if there is a deeper reason for the matching.



that do not (recall the splitting (75)). For integer QH states in the limit of extremely strong magnetic fields, the same viscosity was obtained, up to a factor 2, from a bulk integral (79) of a *boundary* current. We also considered fractional QH states, for which the factor 2 had to replaced by other coefficients with otherwise identical conclusions. As explained in the manuscript, such mismatches should not be seen as paradoxes compared with earlier works on the Hall viscosity, since the splitting between current terms and AB terms in the Berry phase (50) generally has nothing to do with the splitting of quantomorphisms in metric and potential deformations. What is remarkable instead is that the Berry phase (50) contains an extensive gauge-invariant piece that just happens to be proportional to the Hall viscosity.

These results open the door to a number of follow-up questions. First, a detailed study of linear response is in order, both for non-interacting QH states and for their fractional peers, to clarify the relation between the integral viscosity formula (79) and the more standard link between viscosity and stresses [83, §2]. One could also imagine going beyond the leading order of adiabatic linear response by evaluating the (functional) quantum metric associated with adiabatic planar deformations [84]. Indeed, it is the *metric* of parameter space, rather than the Berry curvature, that eventually justifies the use of hyperbolic geometry in the context of the Hall viscosity [20].

A thornier issue is that the deformations defined here acted both on the metric and on the potential (see eq. (37)), calling for a modification that would allow metric and potential to transform independently. For example, densities and currents of QH droplets in arbitrary (disordered) potentials were studied in [85,86], providing formulas that can presumably be used to deduce Berry phases due to potential deformations alone. Combining this with compensating quantomorphisms would likely yield deformations of the metric alone, as was achieved independently in [27, 36] for fractional QH states viewed as conformal blocks. The link between these approaches seems tenuous and requires clarification, not to mention their relation with earlier works on infinitesimal deformations in the lowest Landau level [12–16]. We intend to investigate this in the near future; see also [78].

Finally, it goes without saying that our predictions call for experiments. Even in the simplest 1D quantum wires treated here as toy models, observing the phases (16) or the corresponding adiabatic response would be fascinating. This may require a reformulation of our formalism in terms of lattice models, along the lines of what was done *e.g.* in conformal field theories with a Floquet drive [87,88]. Even more importantly, implementing 2D planar deformations in QH simulators could provide the striking measure of the Hall viscosity that is still lacking at the moment, despite recent breakthroughs in graphene [38]. This could again be done either on a lattice or in the continuum [42, 43]. Our formalism thus paves the way for a number of new geometric observations in mesoscopic systems, with conceptual implications for topological phases in general.

# Acknowledgments

We are grateful to Laurent Charles for many enlightening discussions on quantomorphisms; to Hansueli Jud and Clément Tauber for introducing us to the Hall viscosity; to Giandomenico Palumbo, Nicolas Regnault, Shinsei Ryu and Jean-Marie Stéphan for insightful comments on the initial preprint; and finally to Mathieu Beauvillain, Nathan Goldman, Bastien Lapierre, Per Moosavi and Marios Petropoulos for collaboration on related subjects. Many thanks also to the organizers of the workshop 'Mathematical Aspects of Quantum Phases of Matter' (held in Będlewo, Poland in July 2021), where a preliminary version of the results of this paper was presented.

**Funding information**    The work of B.O. is supported by the European Union's Horizon 2020 research and innovation programme under the Marie Skłodowska-Curie grant agreement No. 846244. B.O and B.E. are also supported by the ANR grant *TopO* No. ANR-17-CE30-0013-01.

# A   Quantomorphisms on closed surfaces

This appendix accompanies sec. 3.3 and provides a self-contained review of the action of (certain) area-preserving diffeomorphisms on closed, oriented surfaces such as spheres or tori. In the context of geometric quantization (see [44, 45, 70, 71]), the action that we consider coincides with automorphisms of prequantum line bundles, also known as prequantum operators or *quantomorphisms* [89]. A summary of the main points is as follows:

- The Hilbert space of a charged particle coupled to a background U(1) gauge field **A** is a space of sections for a Hermitian line bundle $L$ on some base manifold $\Sigma$, with compatible connection $\nabla$. This geometric data $(L, \nabla)$ is fully captured by (and can be reconstructed from) the knowledge of all holonomies of **A**.

- Not all area-preserving diffeomorphisms can be lifted to unitary quantomorphisms acting on the Hilbert space: only *holonomy-preserving* ones can. The corresponding transformation of wave functions is unique up to a global phase, and can be constructed explicitly using parallel transport. When the base space $\Sigma$ is the plane, this action reduces to eq. (35) since all area-preserving maps also preserve holonomies. This is *not* so on generic surfaces (*e.g.* the torus), in which case eq. (35) only holds locally.

- The Lie algebra of the group of holonomy-preserving diffeomorphisms consists of all Hamiltonian vector fields, that is, vector fields $\boldsymbol{v}$ for which the one-form $\iota_{\boldsymbol{v}}\mathbf{B}$ is exact, where $\mathbf{B} = \mathrm{d}\mathbf{A}$ is the curvature of the connection. The unitary action of any such vector field $\boldsymbol{v}$ is given by the Hermitian operator (40):

$$\mathfrak{u}[\boldsymbol{v}] = -i\nabla_{\boldsymbol{v}} - F_{\boldsymbol{v}}, \qquad \text{with} \qquad \iota_{\boldsymbol{v}}\mathbf{B} = \mathrm{d}F_{\boldsymbol{v}}, \tag{A.1}$$

  where $\nabla_{\boldsymbol{v}}$ is the covariant derivative along $\boldsymbol{v}$ and $F_{\boldsymbol{v}}$ is a stream function (a 'Hamiltonian') for $\boldsymbol{v}$ in the sense of sec. 3.2.

The plan of the appendix is as follows. We begin by recalling the fiber bundle approach to quantum mechanics, where charged wave functions on $\Sigma$ are viewed as sections of a complex line bundle equipped with a connection. We then ask how diffeomorphisms of $\Sigma$ can be implemented unitarily, and show that only Hamiltonian area-preserving maps are allowed, leading to the quantomorphisms defined locally in eq. (35). Finally, we turn to infinitesimal quantomorphisms and recover the earlier expression (40), now derived thanks to bundle geometry. We adopt units such that $q/\hbar = 1$ for simplicity.

We stress that the material of this appendix is not original, and is for the most part a simple translation of the lecture notes [89] from the language of principal U(1) bundles to that of the associated line bundles. We hope to have made it accessible to physicists: the only required background is some familiarity with line bundles, as reviewed *e.g.* in [61, chaps. 9–10]. We also refer to [90] for a pedagogical introduction to geometric quantization that overlaps with our presentation here.

## A.1   Line bundles, connections and holonomies

Let us briefly review the geometric treatment of quantum mechanics, in which wave functions are sections of a Hermitian line bundle [44, 91, 92]. Background electromagnetic fields are then

incorporated thanks to a connection, generally with some non-zero curvature (= magnetic field). This approach is very general, but we focus on the case relevant for the quantum Hall effect, namely a charged particle moving on some 2D surface $\Sigma$. The latter is assumed to be connected, closed (*i.e.* compact and without boundary), and oriented. We endow it with an area form $\boldsymbol{\omega}$, *i.e.* a nowhere-vanishing two-form; this is typically the Riemannian area form $\boldsymbol{\omega} = \sqrt{g}\,\mathrm{d}x \wedge \mathrm{d}y$ associated with a Riemannian metric $g$, but the metric will play no role in the following.

**Line bundles and sections.** How to describe wave functions on $\Sigma$? The starting point is to think of $\Sigma$ as the base space of a complex line bundle $\pi : L \to \Sigma$. Wave functions are sections of this bundle, *i.e.* maps $\Psi : \Sigma \to L : \mathbf{x} \mapsto \Psi(\mathbf{x})$ such that $\pi(\Psi(\mathbf{x})) = \mathbf{x}$ for any point $\mathbf{x}$ in $\Sigma$. To make sense of the fact that wave functions must be square-integrable, we assume that $L$ is endowed with a Hermitian structure. Scalar products of sections thus read

$$\langle \Phi | \Psi \rangle = \int_\Sigma h(\Phi, \Psi)\,\boldsymbol{\omega}\,, \tag{A.2}$$

where $h$ is a Hermitian metric, *i.e.* a collection of Hermitian inner products $h_\mathbf{x}\big(\Phi(\mathbf{x}), \Psi(\mathbf{x})\big)$ on each fiber at $\mathbf{x} \in \Sigma$.

The line bundle of sec. 3.3 was a trivial direct product $L = \Sigma \times \mathbb{C}$ with $\Sigma = \mathbb{R}^2$. Sections were thus functions $\mathbf{x} \mapsto \Psi(\mathbf{x})$ and the Hermitian structure (A.2) was given by $h_\mathbf{x}\big(\Phi(\mathbf{x}), \Psi(\mathbf{x})\big) = \Phi^*(\mathbf{x})\Psi(\mathbf{x})$. In general, however, the line bundle is non-trivial and cannot be written globally as a product manifold. Relatedly, sections cannot be written as mere functions on $\Sigma$. One therefore has to choose some covering of $\Sigma$ by a (finite) collection of open sets $U_a$ labelled by some index $a = 1, 2, \dots$. These sets can always be taken small enough that their preimage $\pi^{-1}(U_a) \cong U_a \times \mathbb{C}$ be a trivial bundle $\pi : \pi^{-1}(U_a) \to U_a$; we then refer to the open covering $\{U_a\}$ as *trivializing*. This can be used to locally write any section as a function. Indeed, choose for each $U_a$ a section $\sigma_a : U_a \to L$ that vanishes nowhere on $U_a$; we refer to this as a choice of *frame* on the line bundle. Given a frame, write any global section as (no implicit summation over $a$!)

$$\Psi(\mathbf{x}) = \psi_a(\mathbf{x})\,\sigma_a(\mathbf{x})\,, \qquad \forall\,\mathbf{x} \in U_a\,, \tag{A.3}$$

with $\psi_a$ some complex function on $U_a$. The frame is said to be *unitary* if $h_\mathbf{x}\big(\sigma_a(\mathbf{x}), \sigma_a(\mathbf{x})\big) = 1$ for all $\mathbf{x} \in U_a$. Note that the choice of frame is not unique: for any unitary frame $\{\sigma_a\}$, one can use local gauge transformations on the $U_a$'s to define a new unitary frame

$$\tilde{\sigma}_a(\mathbf{x}) \equiv e^{-i\alpha_a(\mathbf{x})}\sigma_a(\mathbf{x})\,. \tag{A.4}$$

The corresponding local wave function $\psi_a$ transforms into $\tilde{\psi}_a(\mathbf{x}) = e^{i\alpha_a(\mathbf{x})}\psi(\mathbf{x})$. Also note that, given a unitary frame, wave functions on non-empty overlaps $U_a \cap U_b$ are related by similar gauge transformations: one has $\psi_a(\mathbf{x}) = e^{i\alpha_{ab}(\mathbf{x})}\psi_b(\mathbf{x})$.

**Background field as a connection.** We now wish to couple our quantum system on $\Sigma$ to some background U(1) gauge field. This is achieved by endowing the line bundle $L \to \Sigma$ with a connection $\nabla$ that defines a covariant derivative of sections, *i.e.* a notion of parallel transport along $\Sigma$. The latter is readily written in a familiar form upon using some local trivialization $\{U_a\}$ and some unitary frame $\{\sigma_a\}$: one can then specify the connection on each $U_a$ by a local one-form $\mathbf{A}_a$ such that

$$\nabla_{\boldsymbol{v}}\,\sigma_a \equiv -i\langle \mathbf{A}_a, \boldsymbol{v} \rangle\,\sigma_a\,, \tag{A.5}$$

for any vector field $\boldsymbol{v}$ on $U_a$, with $\langle \mathbf{A}_a, \boldsymbol{v} \rangle \equiv \iota_{\boldsymbol{v}} \mathbf{A}_a = v^j A_{a,j}$ as in eq. (43). Thus the covariant derivative of an arbitrary section $\Psi$, written locally as (A.3), reads

$$\nabla_{\boldsymbol{v}}\Psi = \big(v^j \partial_j \psi_a - i\langle \mathbf{A}_a, \boldsymbol{v} \rangle\,\psi_a\big)\sigma_a\,. \tag{A.6}$$

The term in parentheses on the right-hand side is the familiar form of a covariant derivative, expressed in terms of components $v^j$ and involving the local gauge field $\mathbf{A}_a$.

As usual, we require the connection to be unitary, *i.e.* that the connection one-forms $\mathbf{A}_a$ be real for any unitary frame. (In fact, if $\mathbf{A}_a$ is real in a unitary frame, then it is real in any other unitary frame.) Parallel transport along any curve in $\Sigma$ is thus an isometry: it preserves the Hermitian product (A.2). In particular, parallel transport along a closed, oriented loop $\gamma$ in $\Sigma$ only affects a wave function by an overall phase, namely the *holonomy* of the connection around $\gamma$. From the perspective of sec. 4, this holonomy is nothing but an Aharonov-Bohm phase. It turns out that the full geometric data of the line bundle $L$ with connection $\nabla$ is encoded in such holonomies: two Hermitian bundles with connection are isomorphic if and only if all their holonomies coincide (see *e.g.* [93]).

Note that changing the local frame changes connection one-forms: the local gauge transformations (A.4) affect the gauge fields $\mathbf{A}_a$ defined in (A.5) by changing them into $\tilde{\mathbf{A}}_a = \mathbf{A}_a + \mathrm{d}\alpha_a$. On overlaps where $\psi_a(\mathbf{x}) = e^{i\alpha_{ab}(\mathbf{x})}\psi_b(\mathbf{x})$, one similarly finds $\mathbf{A}_a = \mathbf{A}_b + \mathrm{d}\alpha_{ab}$.

**Magnetic field as a curvature.** Suppose one is given a Hermitian line bundle with some unitary connection $\nabla$. Then the corresponding curvature two-form $\mathbf{B}$ is defined by the commutator of covariant derivatives,

$$\nabla_v \nabla_w - \nabla_w \nabla_v - \nabla_{[v,w]} = -i\mathbf{B}(v,w). \tag{A.7}$$

In terms of a local trivialization with some local frame, this yields the usual magnetic field $\mathbf{B} = \mathrm{d}\mathbf{A}_a$ on $U_a$. We shall assume that $\mathbf{B}$ is uniform: given the area form $\boldsymbol{\omega}$ on $\Sigma$, we let $\mathbf{B} = B\boldsymbol{\omega}$ so that $\int_\Sigma \mathbf{B} = B \times \mathrm{area}(\Sigma)$, and assume $B > 0$ for definiteness.

There is little more to be said about uniform magnetic fields when $\Sigma$ is the plane, as was the case in the main text. However, subtleties occur on closed manifolds, where the two-form $\mathbf{B}$ is not exact (as it would otherwise violate Stokes's theorem) and is subject to Dirac's quantization condition

$$\frac{1}{2\pi} \int_\Sigma \mathbf{B} \in \mathbb{Z}. \tag{A.8}$$

This integer is the Chern number of the line bundle. As usual, the bundle is non-trivial if the Chern number is non-zero.

## A.2 Lifting diffeomorphisms to line bundles

Having recalled the bundle picture of quantum mechanics, we now use it to study deformations of the base manifold. Namely, given a diffeomorphism $\boldsymbol{g} : \Sigma \to \Sigma$, we wish to define a corresponding unitary operator acting on the Hilbert space of square-integrable sections $\Psi : \Sigma \to L$. The most naive solution is to send $\Psi$ on its pullback $\boldsymbol{g}^*\Psi = \Psi \circ \boldsymbol{g}$, possibly adding a prefactor as in eq. (11) to account for the fact that a wave function is a half-density.[14] But this does not quite make sense: $(\boldsymbol{g}^*\Psi)(\mathbf{x})$ lies in the fiber $L_{\boldsymbol{g}(\mathbf{x})}$ instead of $L_\mathbf{x}$, so $\boldsymbol{g}^*\Psi$ is no longer a section of the bundle $\pi : L \to \Sigma$ and fails to be a *bona fide* wave function. On the other hand, $\boldsymbol{g}^*\Psi$ is a section of the pullback bundle $\pi' : \boldsymbol{g}^*L \to \Sigma$, so we need a way to identify these two bundles. This is achieved thanks to a bundle isomorphism, *i.e.* a diffeomorphism $\boldsymbol{G} : L \to \boldsymbol{g}^*L$ such that the following diagram commutes:

$$
\begin{array}{ccc}
L & \xrightarrow{\ \boldsymbol{G}\ } & \boldsymbol{g}^*L \\
{\scriptstyle \pi}\big\downarrow & & \big\downarrow{\scriptstyle \pi'} \\
\Sigma & \xrightarrow[\ \mathrm{id}\ ]{} & \Sigma
\end{array}
$$

---

[14]We will eventually see in eq. (A.12) that $\boldsymbol{g}$ needs to preserve area, so no square root appears in practice.

and whose restriction to each fiber is a linear map. Morally, the isomorphism is just a collection of (invertible) linear maps $G_{\mathbf{x}} : L_{\mathbf{x}} \to L_{g(\mathbf{x})}$, depending smoothly on $\mathbf{x} \in \Sigma$. Provided such a map exists, we may assume without loss of generality that each $G_{\mathbf{x}}$ is unitary. (If $G_{\mathbf{x}}$ is not norm-preserving, it can be rescaled fiberwise $G_{\mathbf{x}} \to \lambda_{\mathbf{x}} G_{\mathbf{x}}$ with some smooth rescaling factor $\lambda : \Sigma \to \mathbb{R}^+$.) Alternatively, one can interpret $G$ as a bundle map covering $g$, in the sense that the following diagram commutes:

$$
\begin{array}{ccc}
L & \xrightarrow{\;G\;} & L \\
{\scriptstyle \pi}\downarrow & & \downarrow{\scriptstyle \pi} \\
\Sigma & \xrightarrow{\;g\;} & \Sigma
\end{array}
$$

The bundle map $G$ thus sends $g^*\Psi$ back to $L$, yielding the following unitary action of $g$ on sections of $L$:

$$
\mathcal{U}[g]\Psi \equiv G^{-1} \circ \Psi \circ g \,. \tag{A.9}
$$

This is the elementary starting point that will eventually yield quantomorphisms.

In practice, the prescription (A.9) is subject to a number of conditions. First, it requires a bundle isomorphism between $L$ and $g^*L$. The latter exists if and only if $g^*L$ and $L$ have the same first Chern class $c_1[L] = c_1[g^*L] \in H^2(\Sigma, \mathbb{Z})$, which here boils down to $g$ being orientation-preserving.[15] A second, much stronger requirement is that $G$ must not affect the coupling of wave functions to the background gauge field. This is to say that the operator (A.9) needs to leave the connection invariant:

$$
\mathcal{U}[g]^\dagger \, \nabla_v \, \mathcal{U}[g] = \nabla_{g_*v} \,, \tag{A.10}
$$

where $v$ is any vector field on $\Sigma$ and $g_*v$ is its pushforward by $g$. Thus the image of the connection $\nabla$ under the bundle isomorphism $G : L \to g^*L$ must be the pullback connection $g^*\nabla$. Equivalently, the bundle map $G : L \to L$ covering $g$ must preserve the connection $\nabla$. Unitary bundle isomorphisms that satisfy these criteria are known as *prequantum bundle automorphisms* or *quantomorphisms* [44,45]. As emphasized in sec. 3.3, magnetic translations of electronic wave functions provide a well known example of such maps in physics.

The key constraint (A.10) limits sharply the set of deformations of $\Sigma$ that admit a well-defined unitary action on charged wave functions. Indeed, $(L, \nabla)$ and $(g^*L, g^*\nabla)$ need to be isomorphic as Hermitian line bundles *with connection*, so all their holonomies must coincide. As a result, a diffeomorphism $g : \Sigma \to \Sigma$ admits a unitary, connection-preserving lift $G : L \to L$ if and only if it is holonomy-preserving: for any cycle $\gamma$ in $\Sigma$,

$$
\mathrm{hol}(\gamma) = \mathrm{hol}(g \circ \gamma) \,. \tag{A.11}
$$

Note that holonomy-preserving maps necessarily preserve area: if the cycle $\gamma$ is the boundary of some domain $S$, then $\mathrm{hol}(\gamma) = B \times \mathrm{area}(S)$ and eq. (A.11) entails the local condition

$$
g^*\mathbf{B} = \mathbf{B} \,. \tag{A.12}
$$

But there is also a global aspect, since eq. (A.11) even holds for loops that are not boundaries. This plays no role on simply connected surfaces such as $\mathbb{R}^2$ or $S^2$, but it does affect other cases. On the torus, for instance, all translations preserve area but only a finite subgroup satisfies (A.11) [94,95]. We return to this example around eqs. (A.16)–(A.17) below.

---

[15]For compact oriented $\Sigma$, $H^1(\Sigma)$ is torsion-free and the *real* first Chern class $[\mathbf{B}] \in H^2(\Sigma, \mathbb{R})$ classifies line bundles. Then $g^*\mathbf{B}$ and $\mathbf{B}$ are cohomologous iff $\int_\Sigma g^*\mathbf{B} = \int_\Sigma \mathbf{B}$, *i.e.* if $g$ preserves orientation.

### A.3 Quantomorphisms from parallel transport

We have just seen that a diffeomorphism can be made to act on the Hilbert space of a charged particle provided it preserves all holonomies. But while the existence of the corresponding unitary action (A.9) is guaranteed, its explicit form in terms of transformed wave functions is lacking. It turns out that this is easily fixed using parallel transport. The key point is that the connection-preserving lift $G$ is fully characterized (up to a global phase) by the fact that it is unitary and commutes with parallel transport in the sense of eq. (A.10). Indeed, pick a reference point $\mathbf{x}_0 \in \Sigma$ and choose a unitary map $G_{\mathbf{x}_0}$ between $L_{\mathbf{x}_0}$ and $L_{g(\mathbf{x}_0)}$; then extend this map smoothly (and uniquely) to all fibers $L_{\mathbf{x}}$ using parallel transport, as follows: given $\mathbf{x} \in \Sigma$, pick *any* curve $\gamma$ from $\mathbf{x}_0$ to $\mathbf{x}$ and let $\mathbf{T}_\gamma$ and $\mathbf{T}'_\gamma = \mathbf{T}_{g(\gamma)}$ denote parallel transport above $\gamma$ in $L$ and $g^*L$, respectively. Since $G$ commutes with parallel transport, one must have $G_{\mathbf{x}} = \mathbf{T}'_\gamma \circ G_{\mathbf{x}_0} \circ \mathbf{T}_\gamma^{-1}$. The latter does not depend on the choice of $\gamma$ (thanks to the condition (A.11)) and it depends smoothly on $\mathbf{x}$, yielding the bundle isomorphism needed for eq. (A.9).

This is the intrinsic, general formulation of the argument for planar quantomorphisms presented in sec. 3.3. Similarly to that simpler case, the resulting operators (A.9) depend on one's choice of $G_{\mathbf{x}_0}$, leading to an unavoidable global phase ambiguity. This was to be expected, since the connection-preserving unitary lift $G$ of a map $g$ is only unique up to a global phase. (Indeed, if $G_1$ and $G_2$ are two lifts of the same $g$, then $G_1^{-1} \circ G_2$ is a gauge transformation that leaves the connection invariant, *i.e.* a global phase.)

**Example 1: the plane.** Let us illustrate this construction of quantomorphisms with two examples. First, on the plane $\mathbb{R}^2$ with $\mathbf{B} = B \, dx \wedge dy$, any area-preserving deformation also preserves all holonomies. In order to fix the map $G_{\mathbf{x}_0}$ in a gauge invariant way, one can use parallel transport along a curve $\gamma_g$ from $\mathbf{x}_0$ to $g(\mathbf{x}_0)$. Given a global frame $\sigma$ corresponding to a global gauge choice $\mathbf{A}$, the lift of $g$ is then

$$G_{\mathbf{x}}\sigma(\mathbf{x}) = e^{i\int_{\gamma_g} \mathbf{A}} e^{i\int_{\mathbf{x}_0}^{\mathbf{x}}(g^*\mathbf{A}-\mathbf{A})}\sigma(g(\mathbf{x})), \tag{A.13}$$

where $\mathbf{x}_0 \in \mathbb{R}^2$ is an arbitrary reference point, and the actual integration path from $\mathbf{x}_0$ to $\mathbf{x}$ is irrelevant since $g^*\mathbf{A}-\mathbf{A}$ is closed. Changing the reference point $\mathbf{x}_0$ and/or the curve $\gamma_g$ yields the same lift up to a constant phase. The corresponding unitary action on wave functions reproduces eq. (35):

$$(\mathcal{U}[g]\psi)(\mathbf{x}) = e^{-i\int_{\gamma_g} \mathbf{A}} e^{i\int_{\mathbf{x}_0}^{\mathbf{x}}(\mathbf{A}-g^*\mathbf{A})}\psi(g(\mathbf{x})). \tag{A.14}$$

As in sec. 3.3, the constant phase $e^{-i\int_{\gamma_g} \mathbf{A}}$ may be dropped in principle, but ensures in practice that the operator (A.14) is independent of the gauge used to define it.

**Example 2: the torus.** Our second example is the group of translations on the torus $\mathbb{T}^2 = \mathbb{R}^2/\mathbb{Z}^2$ with an area form $\boldsymbol{\omega} = dx \wedge dy$ and magnetic field $\mathbf{B} = B\boldsymbol{\omega}$. The latter is quantized according to $B = 2\pi C$, where the integer $C$ is the Chern number of the line bundle. As the bundle is non-trivial in general, several coordinate charts are typically needed to describe its sections. Alternatively, one can use a single chart [95], say $U = \{0 < x < 1, 0 < y < 1\}$, at the cost of working with quasiperiodic wave functions such as

$$\psi(x+1,y) = e^{i\alpha}e^{iBy}\psi(x,y), \qquad \psi(x,y+1) = e^{i\beta}\psi(x,y), \tag{A.15}$$

written here in Landau gauge $\mathbf{A} = B \, x \, dy$ (tantamount to a choice of unitary frame $\sigma$ on $U$). The phases $\alpha$ and $\beta$ are physical observables: they can be measured by holonomies around suitable non-contractible cycles of the torus. Changing $\alpha$ or $\beta$ changes the connection, so

holonomy-preserving diffeomorphisms must preserve area while *also* leaving $\alpha, \beta$ invariant. As a result, only a finite subgroup of translations is holonomy-preserving: it is generated by

$$\mathbf{t}_1(x, y) = (x + 1/C, y), \qquad \mathbf{t}_2(x, y) = (x, y + 1/C). \tag{A.16}$$

The corresponding lifts $\mathbf{T}_i : L \to L$ covering $\mathbf{t}_i : \mathbb{T}^2 \to \mathbb{T}^2$ act on the local frame $\sigma$ as $\mathbf{T}_1 \sigma(x, y) = e^{2\pi i y} \sigma(x + 1/C, y)$ and $\mathbf{T}_2 \sigma(x, y) = \sigma(x, y + 1/C)$, so their unitary action (A.9) is

$$(\mathcal{U}_1 \psi)(x, y) = e^{-2\pi i y} \psi(x + 1/C, y), \qquad (\mathcal{U}_2 \psi)(x, y) = \psi(x, y + 1/C), \tag{A.17}$$

up to a global phase. In particular, one has $\mathcal{U}_1 \mathcal{U}_2 = e^{2\pi i/C} \mathcal{U}_2 \mathcal{U}_1$, which is yet another instance of non-commuting magnetic translations.

## A.4  Hamiltonian vector fields and infinitesimal quantomorphisms

We conclude this appendix by turning to infinitesimal quantomorphisms. In particular, one may wonder what divergence-free vector fields are such that their flow $\boldsymbol{g}_t$, given by eq. (28), preserves all holonomies. The answer turns out to be very simple: the generators of holonomy-preserving diffeomorphisms are *Hamiltonian* vector fields, *i.e.* vector fields $\boldsymbol{v}$ such that $\iota_{\boldsymbol{v}} \mathbf{B}$ be *exact* as opposed to merely closed. Those are the vector fields that admit a stream function.[16]

To prove this, demand that the flow $\boldsymbol{g}_t = \exp(t\boldsymbol{v})$ be holonomy-preserving for all $t$. This is the case provided the holonomy along $\boldsymbol{g}_t(\boldsymbol{\gamma})$ does not depend on $t$, for any loop $\boldsymbol{\gamma}$ in $\Sigma$. But $\boldsymbol{g}_t(\boldsymbol{\gamma}) - \boldsymbol{\gamma}$ is the boundary of the worldsheet traced by $\boldsymbol{g}_s(\boldsymbol{\gamma})$ as $s$ varies from 0 to $t$, so the condition boils down to the absence of magnetic flux through this worldsheet. Infinitesimally, this means $\oint_{\boldsymbol{\gamma}} \iota_{\boldsymbol{v}} \mathbf{B} = 0$ for any cycle $\boldsymbol{\gamma}$, which is to say that $\iota_{\boldsymbol{v}} \mathbf{B} = \mathrm{d} F$ is exact in terms of some stream function $F$.

Now pick a Hamiltonian vector field $\boldsymbol{v}$ with some stream function $F_{\boldsymbol{v}}$. What is the corresponding Hermitian operator (7) obtained by considering infinitesimal quantomorphisms? We already wrote the planar result in eq. (40) above, but it is worth revisiting it here from the bundle perspective. Accordingly, consider time-dependent deformations $\boldsymbol{g}_t$ such that $\boldsymbol{g}_0 = e$ is the identity map on $\Sigma$, while $\boldsymbol{g}_1 \equiv \boldsymbol{g}$ is some holonomy-preserving diffeomorphism. (This is really an isotopy from the identity to $\boldsymbol{g}$.) Then lift each $\boldsymbol{g}_t$ to a bundle map via parallel transport. To do this, start by fixing some point $\mathbf{x} \in \Sigma$ and some time $t$, and define a parallel transport map $T_{\mathbf{x},t} : L_{\mathbf{x}} \to L_{\boldsymbol{g}_t(\mathbf{x})}$, which is a linear (and in fact unitary) isomorphism between $L_{\mathbf{x}}$ and $L_{\boldsymbol{g}_t(\mathbf{x})}$. Then build a unitary bundle map

$$\mathbf{T}_t : L \to L : \mathbf{X} \to \mathbf{T}_t(\mathbf{X}) \equiv T_{\pi(\mathbf{X}),t}(\mathbf{X}), \tag{A.18}$$

so that the following diagram commutes:

$$
\begin{array}{ccc}
L & \xrightarrow{\ \mathbf{T}_t\ } & L \\
\pi \downarrow & & \downarrow \pi \\
M & \xrightarrow{\ \boldsymbol{g}_t\ } & M
\end{array}
$$

The only issue at this stage is that parallel transport is not connection-preserving: the bundle map (A.18) does not satisfy the constraint (A.10), since acting with $\mathbf{T}_t$ changes the connection to $\nabla_t = \nabla - i\boldsymbol{\beta}_t$ with a real-one form $\boldsymbol{\beta}_t$ that satisfies $\frac{\mathrm{d}\boldsymbol{\beta}_t}{\mathrm{d}t} = \boldsymbol{g}_t^* \iota_{\boldsymbol{v}_t} \mathbf{B}$. This can be compensated by a gauge transformation, *i.e.* a bundle automorphism covering the identity, if and only if $\boldsymbol{\beta}_t$ is exact. Thus we recover the fact that $\boldsymbol{v}_t$ has to be a Hamiltonian vector field, meaning that

---

[16]When the first De Rham cohomology $H^1(\Sigma, \mathbb{R})$ vanishes, all closed forms are exact, so all divergence-free vector fields are Hamiltonian. This happens on the plane and the sphere. But higher genus surfaces admit divergence-free vector fields that are *not* Hamiltonian, such as $\partial_x$ and $\partial_y$ on the torus. Recall footnotes 10–11.

there exists a smooth function $F_t : \Sigma \to \mathbb{R}$ such that $\iota_{v_t} \mathbf{B} = dF_t$. Provided this holds, one can build a connection-preserving lift

$$\boldsymbol{G}_t \equiv \mathbf{T}_t \circ e^{i\theta_t}, \qquad \text{with} \qquad \frac{d\theta_t}{dt} = \boldsymbol{g}_t^* F_t, \tag{A.19}$$

where $e^{i\theta_t}$ stands for fiberwise multiplication by $e^{i\theta_t(x)}$. The flow of the Lie algebra operator (7) follows as an immediate corollary. Indeed, for an autonomous vector field $\boldsymbol{v}$ with stream function $F_v$, the compensating gauge transformation is $\theta_t(x) = t F_v(x)$ since the Hamiltonian $F_v$ is conserved along the flow $t \to \boldsymbol{g}_t(\mathbf{x})$. This finally yields the so-called Kostant-Souriau prequantum operator [44, 75]

$$\mathfrak{u}[\boldsymbol{v}] = -i\nabla_{\boldsymbol{v}} - F_{\boldsymbol{v}}, \tag{A.20}$$

where the two terms on the right-hand side respectively generate parallel transport ($\nabla_{\boldsymbol{v}}$) and a gauge correction ($F_{\boldsymbol{v}}$). The same interpretation led to planar quantomorphisms (35), and it was confirmed in (40) for Lie algebra operators.

# B Cocycle contribution to the Berry phase

This short technical appendix completes the proof of the Berry phase (49) in section 4.2. Our goal is to evaluate the contribution of the cocycle (39) in the Berry phase (10). Using (39) and the fact that $\boldsymbol{g}_t$ preserves area for all $t$, we write the cocycle needed for eq. (10) as

$$\frac{\hbar}{q} C(\boldsymbol{g}_\tau, \boldsymbol{g}_t^{-1}) = \int_{\gamma_{\boldsymbol{g}_\tau \circ \boldsymbol{g}_t^{-1}}} \boldsymbol{g}_t^{-1*} \mathbf{A} - \int_{\gamma_{\boldsymbol{g}_\tau}} \boldsymbol{g}_t^{-1*} \mathbf{A} - \int_{\gamma_{\boldsymbol{g}_t^{-1}}} \boldsymbol{g}_\tau^* \boldsymbol{g}_t^{-1*} \mathbf{A}. \tag{B.1}$$

Let us assume for simplicity that the path $\gamma_{\boldsymbol{g}}$ is the straight line from $\mathbf{x}_0$ to $\boldsymbol{g}(\mathbf{x}_0)$. Then the time derivative of the first term in (B.1) is

$$\partial_\tau|_{\tau=t} \int_{\gamma_{\boldsymbol{g}_\tau \circ \boldsymbol{g}_t^{-1}}} \boldsymbol{g}_t^{-1*} \mathbf{A} = (\iota_{v_t} \boldsymbol{g}_t^{-1*} \mathbf{A})(\mathbf{x}_0) = -\mathbf{A}_{\boldsymbol{g}_t^{-1}(\mathbf{x}_0)}(\partial_t \boldsymbol{g}_t^{-1}(\mathbf{x}_0)), \tag{B.2}$$

where $\boldsymbol{v}_t$ is the logarithmic derivative (28). This is in fact the result quoted below eq. (56) of the main text, so we now need to prove that the remaining derivatives of (B.1) cancel out. To do this, consider first the second term in (B.1), whose time derivative is

$$-\partial_\tau|_{\tau=t} \int_{\gamma_{\boldsymbol{g}_\tau}} \boldsymbol{g}_t^{-1*} \mathbf{A} = \int_{\gamma_{\boldsymbol{g}_t}} \frac{d}{dt} \boldsymbol{g}_t^{-1*} \mathbf{A} = \int_{\gamma_{\boldsymbol{g}_t}} \boldsymbol{g}_t^{-1*} \mathcal{L}_{w_t} \mathbf{A}, \tag{B.3}$$

up to an irrelevant total time derivative, where $\boldsymbol{w}_t \equiv -\boldsymbol{g}_t^* \boldsymbol{v}_t$ is the velocity (28) of the *inverse* flow $\boldsymbol{g}_t^{-1}$. Since the form $\boldsymbol{g}_t^{-1*} \mathcal{L}_{w_t} \mathbf{A} = -\mathcal{L}_{v_t} \boldsymbol{g}_t^{-1*} \mathbf{A}$ is closed, the actual integration path does not matter and one can rewrite (B.3) as

$$-\partial_\tau|_{\tau=t} \int_{\gamma_{\boldsymbol{g}_\tau}} \boldsymbol{g}_t^{-1*} \mathbf{A} = -\int_{\mathbf{x}_0}^{\boldsymbol{g}_t(\mathbf{x}_0)} \mathcal{L}_{v_t} \boldsymbol{g}_t^{-1*} \mathbf{A}. \tag{B.4}$$

Finally, the time derivative of the last term in (B.1) is

$$-\partial_\tau|_t \int_{\gamma_{\boldsymbol{g}_t^{-1}}} \boldsymbol{g}_\tau^* \boldsymbol{g}_t^{-1*} \mathbf{A} = -\int_{\gamma_{\boldsymbol{g}_t^{-1}}} \boldsymbol{g}_t^* \mathcal{L}_{v_t} \boldsymbol{g}_t^{-1*} \mathbf{A} = -\int_{\boldsymbol{g}_t(\gamma_{\boldsymbol{g}_t^{-1}})} \mathcal{L}_{v_t} \boldsymbol{g}_t^{-1*} \mathbf{A} = \int_{\mathbf{x}_0}^{\boldsymbol{g}_t(\mathbf{x}_0)} \mathcal{L}_{v_t} \boldsymbol{g}_t^{-1*} \mathbf{A}, \tag{B.5}$$

which exactly cancels the second term (B.4), confirming that $\frac{\hbar}{q} \partial_\tau C(\boldsymbol{g}_\tau, \boldsymbol{g}_t^{-1})$ coincides with (B.2) up to an irrelevant total time derivative that was neglexted in (B.3).

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
