# Peer review of "Adiabatic Deformations of Quantum Hall Droplets"

_SciPost Physics, doi:SciPost Phys. 15, 159 (2023)_

## Round 2 · Referee Report · Anonymous (Referee 4) · 2023-5-31

Strengths

Elegant result for the Berry phase for general geometric deformations.

Clear and eloquent presentation.

Weaknesses

Relation to previous calculations of the Hall viscosity not clear.

Claims on the possible measurement of the Berry phase seem a bit far-fetched.

Report

The manuscript addresses the response of many body quantum systems, specifically quantum Hall fluids, to adiabatic geometry deformations, and the resulting Berry phase. The Authors obtain an elegant expression, composed of two parts, one containing the current, which mainly stems from the edge region, and the second representing a bulk Aharonov-Bohm phase. They discuss the relation of their result to the Hall viscosity in the case of specific deformations (more on this below). This is a very nice result, which goes beyond previous discussions in the field (limited to restricted deformations). The presentation is quite eloquent and clear. I therefore believe this manuscript could be suitable for publication in SciPost Physics. However, several points need to be addressed first:
My main concern is the discrepancy between the Berry phase found in this work and the corresponding Hall viscosity calculations in the literature. Perhaps the most relevant here is Ref. [26], which finds the Berry curvature (from which the Berry phase of course follows) for linear transformations (which are enough for calculating the viscosity) in a plane (rather than torus) geometry, similarly to the current manuscript. Sec. II.A shows how by careful calculation the super-extensive contributions cancel out, and the correct Hall viscosity arises. Why does the result in the manuscript seem to be different?
Let me also note that Ref. [26] considers more general states (beyond integer and Laughlin quantum Hall states), and work finds that in general the Hall viscosity is related to the orbital spin. Let me suggest the Authors apply their formula to cases such as neutral p+ip superfluids or the Moore-Read state, and compare the Berry phase they find with the general result for the Hall viscosity.
Another issue I would like to mention is the Author's suggestion that the Berry phase might offer a better way to extract the Hall viscosity than current measurement schemes. Although electron hydrodynamic experiments are challenging, extracting the many body Berry phase seems much harder. Could the Authors comment on that, or else modify their statement?

Requested changes

See above.

---

## Round 2 · Referee Report · Anonymous (Referee 3) · 2023-6-3

Strengths

Explicit `ab-initio' calculation of the effects of area-preserving diffeomorphisms on the Landau Level quantum states, obtaining the Berry phase and other effects.

Weaknesses

This is the first step in a rather long project of extracting universal features directly from the many-body problem. Results do not yet fully compare with the results of the effective field theory approach, which has received confirmation and cannot be questioned, but rather complemented.

Report

I think this is a very honest approach to explicitly calculating universal features of the Laughlin Hall states (primarily integer), in particular by using their area-preserving symmetry. Most of these features have been already obtained by a variety of methods, sometimes direct, sometimes indirect by guessing the effective field theory. This is nonetheless room for better checking the existing effective theory/conformal theory of edge excitations and for obtaining further geometric features, especially for what concerns the bulk excitations, which are currently being investigated by many authors. I would recommend the acceptance of the paper as it is.

Requested changes

Please add the following papers to the references regarding the W-infinity symmetry: Cappelli and Maffi, arXiv:2103.04163, arXiv:1801.03759. These papers are more recent than those already referred to and discuss explicit properties of Laughlin states in the same spirit as the present work.

---

## Round 3 · Referee Report · Anonymous (Referee 1) · 2023-8-22

Strengths

As before:
Elegant result for the Berry phase for general geometric deformations.
Clear and eloquent presentation.

Weaknesses

None (previous ones addressed).

Report

In the response letter and resubmitted manuscript the Authors have, in my opinion, properly addressed previous comments by both Referees. Hence, based on my previous evaluation, I now recommend the publication of the manuscript in SciPost Phys.

---

## Round 3 · Referee Report · Anonymous (Referee 2) · 2023-8-28

Strengths

As in my first report: Explicit `ab-initio' calculation of the effects of area-preserving diffeomorphisms on the Landau Level quantum states.

Weaknesses

As in my first report: the issue is not to criticize/correct established effective field theory results, but to complement them by explicit direct calculations.

Report

In my first report, I said that the paper could be published, only some references were missing. They have been added, and the requests by the other referee have been satisfied. So I support the publication.

---

## Round 3 · Author Response

We thank the referees for their thoughtful comments and careful reading of the manuscript. We have now implemented the requested changes: details are provided in the list below.

---

## Round 3 · List of Changes

Warnings issued while processing user-supplied markup:

  • Inconsistency: plain/Markdown and reStructuredText syntaxes are mixed. Markdown will be used.
    Add "#coerce:reST" or "#coerce:plain" as the first line of your text to force reStructuredText or no markup.
    You may also contact the helpdesk if the formatting is incorrect and you are unable to edit your text.

1. The main concern of referee #1 was "the discrepancy between the Berry phase found in [our] work and the corresponding Hall viscosity calculations in the literature." As explained in the manuscript, there is no discrepancy since what we compute is not Hall viscosity, despite striking similarities. This was already stated in the original paper, but we have now emphasized the point even more by adding several sentences throughout the text:

A. In the introduction (bottom of page 4), we've added the sentence "In this sense, there was no reason for the Berry phase (1) to be related to viscosity at all; it just so happens that its extensive piece is proportional to the Hall viscosity," along with the footnote "The proportionality could have been guessed on geometric grounds: the parameter space for linear maps (2) is a hyperbolic plane, which has a unique SL(2, R)-invariant Berry curvature up to normalization."

B. Below eq. (75), we've split an earlier sentence in two in order to give more details. The sentence now reads: "The factorization between a and b pieces is manifest, respectively corresponding to deformations of the metric and the (slowly varying) confining potential in the Hamiltonian (36). The resulting Berry phase (49) consists of two parts, respectively involving expectation values of (a†a + aa†) and (b†b + bb†)."

C. Below eq. (78), we've stressed the distinction between our Berry phases and the Hall viscosity by adding the following sentences: "One should not be concerned about this discrepancy: there is no inherent reason for the current portion of the Berry phase (50) to be related to viscosity. The fact that the corresponding Berry curvatures match up to normalization is simply a result of the unique invariant area form on a hyperbolic plane (recall footnote 1). Moreover, it is important to note that the distinction between a and b contributions in the operator (75) is specific to linear deformations (2), and does not align with the separation of (50) into current and AB contributions. Identifying the part of the Berry phase (50) arising solely from metric deformations would pose a significant challenge, extending beyond the scope of the quantomorphisms studied here."

D. Finally in the conclusion: "As explained in the manuscript, such mismatches should not be seen as paradoxes compared with ear- lier works on the Hall viscosity, since the splitting between current terms and AB terms in the Berry phase (50) generally has nothing to do with the splitting of quantomorphisms in metric and potential deformations. What is remarkable instead is that the Berry phase (50) contains an extensive gauge-invariant piece that just happens to be proportional to the Hall viscosity."

2. Again following comments by referee #1, the discussion of fractional quantum Hall states has been extended beyond Laughlin, and now includes any isotropic fractional wave function in the LLL (including Moore-Read at filling 1/2). Here we refer to the text surrounding eqs. (79)-(84) for details; it is essentially identical to the text of the previous version, save for a slight re-ordering and an emphasis on the generality of the result. This includes again a sentences that stresses the distinction between our Berry phases and the Hall viscosity, namely: "A similar mismatch occurs, for instance, in the Moore-Read state at filling $\nu=1/2$, where [27] predicts a Hall viscosity $\eta_H=\frac{3}{2}\hbar􏰝/(8\pi\ell_B^2)$ while our universal current Berry curvature (83) predicts a response coefficient $\frac{1}{2}\hbar􏰝/(8\pi\ell_B^2)$. As previously mentioned, this mismatch is not contradictory; instead, it is a distinguishing characteristic. Hall viscosity is a specific response to pure metric deformations, whereas our current Berry curvature (83) encompasses both metric and potential perturbations, and fundamentally does not have any direct connection to viscosity."

3. The last issue raised by referee #1 was the "suggestion that the Berry phase might offer a better way to extract the Hall viscosity than current measurement schemes. (...) Could the Authors comment on that, or else modify their statement?" This is a fair point, so we've now clarified our proposal to measure Berry phases produced by quantomorphisms by adding a sentence in the introduction (below fig. 2): When writing about quantum geometry, "The latter will be crucial in practice, as the likeliest avenue to observe the effects of the phase (1) is to study adiabatic linear response in quantum simulators, where the high degree of control over microscopic details may help overcome issues of decoherence and disorder." The key point here is the emphasis on quantum simulators (as opposed to actual condensed matter samples), where one may hope to control the microscopic Hamiltonian enough to actually implement time-dependent quantomorphisms of the kind described in our paper; we're currently discussing this with researchers working on topological photonics, but this is very much in the earliest stages so it's difficult to say anything more definite.

  1. As required by referee #2, we've added citations to arXiv:2103.04163 and arXiv:1801.03759.

---

## Editorial Decision

published